# Enhancing Diffusion-based Unrestricted Adversarial Attacks via Adversary Preferences Alignment

**Kaixun Jiang[1], Zhaoyu Chen[1*], Haijing Guo[2], Jinglun Li[1], Jiyuan Fu[2],**
**Pinxue Guo[1], Hao Tang[3], Bo Li[4], Wenqiang Zhang[1,2*]**
[1]College of Intelligent Robotics and Advanced Manufacturing, Fudan University
[2]Shanghai Key Lab of Intelligent Information Processing,
College of Computer Science and Artificial Intelligence, Fudan University
[3]School of Computer Science, Peking University    [4]vivo Mobile Communication Co., Ltd.

## Abstract

Preference alignment in diffusion models has primarily focused on benign human preferences (e.g., aesthetic). In this paper, we propose a novel perspective: framing unrestricted adversarial example generation as a problem of aligning with adversary preferences. Unlike benign alignment, adversarial alignment involves two inherently conflicting preferences: visual consistency and attack effectiveness, which often lead to unstable optimization and reward hacking (e.g., reducing visual quality to improve attack success). To address this, we propose APA (Adversary Preferences Alignment), a two-stage framework that decouples conflicting preferences and optimizes each with differentiable rewards. In the first stage, APA fine-tunes LoRA to improve visual consistency using rule-based similarity reward. In the second stage, APA updates either the image latent or prompt embedding based on feedback from a substitute classifier, guided by trajectory-level and step-wise rewards. To enhance black-box transferability, we further incorporate a diffusion augmentation strategy. Experiments demonstrate that APA achieves significantly better attack transferability while maintaining high visual consistency, inspiring further research to approach adversarial attacks from an alignment perspective. Code is available at https://github.com/deep-kaixun/APA.

## 1 Introduction

Preference alignment, adapting pre-trained diffusion models [60] for diverse human preferences, is increasingly prominent in image generation. This typically involves modeling human preferences with explicit reward models or pairwise data [69], then updating model policies via reinforcement learning [3, 21] or backpropagation with differential reward [12, 57, 41]. However, current research largely centers on benign human preferences like aesthetics and text-image alignment (Figure 1(a)), malicious adversary preferences alignment, where security researchers use diffusion models to create unrestricted adversarial examples [10] has received limited attention. These examples are vital for

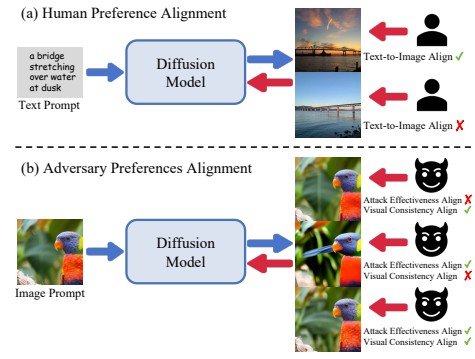

(a) Human Preference Alignment

(b) Adversary Preferences Alignment

Figure 1: Comparison of Human Preference Alignment and Adversary Preferences Alignment.

---

*indicates the corresponding authors.

39th Conference on Neural Information Processing Systems (NeurIPS 2025).

assessing the adversarial robustness of deep learning models. Adversaries, as depicted in Figure 1(b), primarily seek two preferences: 1) Visual consistency: Ensuring that generated images have minimal, semantically negligible differences from the original. 2) Attack effectiveness: Achieving high transferable attack performance, where adversarial examples generated from a surrogate model fool black-box target models [16, 77].

Aligning with such adversarial preferences presents two major challenges. First, preference data is unavailable. Existing diffusion-based attacks [10, 7] build on the idea of traditional $L_p$ attacks [16, 52], adapting the strategy of adding optimized perturbations from the pixel space to the latent space. However, latent spaces in diffusion models are highly sensitive—even slight perturbations can result in severe semantic drift. This makes it infeasible to obtain stable, preference-consistent adversarial examples for pairwise data collection, rendering traditional preference optimization techniques like DPO [69] or unified reward modeling inapplicable. Second, these preferences are inherently in conflict. Joint optimization with reward weighting often results in reward hacking [22], (e.g., one shortcut to improving attack success is to reduce visual consistency), leading to unstable or degenerate solutions (Figure 5(b)).

To address this, we introduce APA (Adversary Preferences Alignment), a novel two-stage framework that separates and sequentially optimizes the adversary preferences using direct backpropagation with differentiable rewards. Specifically: 1) Visual Consistency Alignment: We use a differentiable visual similarity metric as a rule-based reward and perform policy updates by fine-tuning the diffusion model's Low-Rank Adaptation (LoRA) parameters [29]. This stage encodes the input image's structure into the model's generation space, forming a visually stable foundation for downstream attack optimization. 2) Attack Effectiveness Alignment: We optimize either the image latent or the prompt embedding based on feedback from a white-box surrogate classifier. This process uses dual-path attack guidance (both trajectory-level and step-wise dense rewards) to align with the adversary's attack preference. To prevent overfitting to the surrogate, we introduce a diffusion augmentation strategy that aggregates gradients from intermediate steps to increase diversity, thereby improving black-box transferability. Our framework explicitly decouples these conflicting preferences to mitigate reward hacking, enabling more controllable and scalable adversary preferences alignment. Our contributions are summarized as follows:

- We are the first to transform unrestricted adversarial attacks into adversary preferences alignment (APA) and propose an effective two-stage APA framework.

- Our APA framework decouples adversary preferences into two sequential stages which include LoRA-based visual consistency alignment using a rule-based visual similarity reward and attack effectiveness alignment guided by dual-path attack guidance and diffusion augmentation.

- APA achieves state-of-the-art transferability against both standard and defense-equipped models while preserving high visual consistency. Our framework is flexible and scalable, supporting various diffusion models, optimization parameters, update strategies, and downstream tasks.

## 2 Related Work

**Unrestricted Adversarial Attacks.** Unrestricted adversarial attacks address key limitations of traditional $L_p$ attacks, which apply pixel-level perturbations that are often perceptible due to distribution shifts from clean images [33] and are increasingly countered by existing defenses [55, 64]. Instead, unrestricted attacks generate more natural examples by subtly modifying the semantic content of the original images. Early approaches focused on single-type semantic perturbations, including shape [75, 1], texture [58, 36], and color [10] manipulations. Shape-based methods use deformation fields to induce structural changes; texture-based methods modify image texture or style—for example, DiffPGD [79] adds $L_p$ perturbations in pixel space followed by diffusion-based translation, thus falling into this category. Color-based attacks (e.g., SAE [28], ReColorAdv [35], ACE [83]) adjust hue, saturation, or channels to improve visual naturalness, often at the cost of transferability. However, these methods typically optimize a single semantic factor, limiting their generality and expressiveness. Recent efforts leverage the latent space of generative models to produce more flexible adversarial examples. In particular, diffusion models have been adapted for this purpose [10, 7, 56, 13, 9]. For instance, ACA [10] employs DDIM inversion and skip gradients to optimize latent representations. Nonetheless, due to the sensitivity of latent space manipulations, existing approaches often struggle to preserve the visual semantics of the original input. In contrast, our method is the first to approach unrestricted adversarial attacks from a preference alignment perspective. The key idea lies

in alignment-driven modeling: we reframe adversarial attack generation as a preference alignment problem and propose a two-stage framework that decouples the conflicting objectives of visual consistency and attack effectiveness, enabling more stable and controllable generation.

**Alignment of Diffusion Models.** Human preference alignment for diffusion models aims to optimize the pretrained diffusion model based on human reward. Current methods adapted from large language models (LLMs) include reinforcement learning (RL) [3, 21, 6, 37], direct preference optimization (DPO) [69], and direct backpropagation using differentiable rewards (DR) [12, 57, 41]. RL approaches frame the denoising process as a multi-step decision-making process, often using proximal policy optimization (PPO)[62] for fine-tuning. Although flexible, RL is inefficient and unstable when managing conflicting rewards like visual consistency and attack effectiveness. DPO, which avoids reward models by ranking outputs with the Bradley-Terry model, faces challenges in adapting to adversarial preference alignment (APA), given the difficulty of obtaining high-quality adversarial examples for ranking. DR is efficient, using gradient-based optimization with differentiable rewards. To ensure flexibility and stability, we propose a two-stage APA framework built on DR. By decoupling conflicting objectives into differentiable rewards, it effectively addresses the challenges of multi-preferences alignment.

## 3 Preliminary

**Unrestricted Adversarial Example (UAE).** Given a clean image $x$, considering both visual consistency and attack effectiveness, the optimization objective for UAE $x_{adv}$ can be expressed as:

$$\max_{x_{adv}} f_{\phi'}(x_{adv}) \neq y, s.t. \ x_{adv} \text{ is naturally similar to } x, \tag{1}$$

where $y$ denotes the label of $x$, and $f_{\phi'}(\cdot)$ represents target models for which gradients are inaccessible for direct optimization. Since natural similarity cannot be enforced via $L_p$ norms as in perturbation-based attacks [52, 16], unrestricted adversarial attacks must search for optimal adversarial examples in both conflicting optimization spaces.

**Latent Diffusion Model (LDM).** LDM [60] is a latent variable generative model trained on large-scale image-text pairs, relying on an iterative denoising mechanism. During training, the denoising model $\epsilon_\theta(\cdot)$ is trained by minimizing the variational lower bound loss function, typically using a UNet to predict the noise added to the original data:

$$\min_{\epsilon_\theta} \quad E_{t\sim[1,T],\epsilon\sim\mathcal{N}(0,\mathbf{I})} \left\| \epsilon - \epsilon_\theta \left( z_t, t, c \right) \right\|^2, \tag{2}$$

where $t$ denotes the timestep, $T$ denotes the total number of timesteps, and $\epsilon$ is the random noise sampled from $\mathcal{N}(0,\mathbf{I})$. The latent variable $z_t$ is generated by adding noise to $z_0$ over $t$ steps, where $z_0$ is the latent representation of the original input $x$ obtained through encoder $\mathcal{E}(\cdot)$, i.e., $\mathcal{E}(x) = z_0$. The diffusion process is defined as $q(z_t|z_0) = \sqrt{\bar{\alpha}_t} \cdot z_0 + \sqrt{1 - \bar{\alpha}_t} \cdot \epsilon$, where $\sqrt{\bar{\alpha}_t}$ is a hyperparameter that controls the level of noise added at each timestep $t$ [27]. $c$ denotes the conditioning text. During inference, we typically sample $z_T$ from $\mathcal{N}(0,\mathbf{I})$, and then use the DDIM denoising [65] to iteratively denoise $z_T$. The iterative denoising process can be expressed as:

$$z_{t-1} = \sqrt{\bar{\alpha}_{t-1}} \left( \frac{z_t - \sqrt{1-\bar{\alpha}_t}\epsilon_\theta(z_t,t,c)}{\sqrt{\bar{\alpha}_t}} \right) + \sqrt{1 - \bar{\alpha}_{t-1}}\epsilon_\theta \left( z_t, t, c \right). \tag{3}$$

After $T$ steps of DDIM denoising, the resulting $\bar{z}_0$ is decoded into pixel space via a decoder $\mathcal{D}(\cdot)$, generating an image that matches the condition $c$. For tasks with a given reference image (e.g. image editing), $z_T$ is typically not sampled from random noise, instead, it is obtained through DDIM Inversion [65] based on the reference image $x$:

$$z_t = \sqrt{\bar{\alpha}_t} \left( \frac{z_{t-1} - \sqrt{1-\bar{\alpha}_{t-1}} \ \epsilon_\theta(z_{t-1},t,c)}{\sqrt{\bar{\alpha}_{t-1}}} \right) + \sqrt{1 - \bar{\alpha}_t} \ \epsilon_\theta \left( z_{t-1}, t, c \right), \tag{4}$$

where initial $z_0$ denotes the latent of reference image $x$. Through whole DDIM inversion, i.e., iteratively using Eq. 4 $T$ times, we obtain $z_T$, which preserves information of $x$.

## 4 Adversary Preferences Alignment Framework

Existing works [10] leverage the natural image generation capabilities of diffusion models to generate unrestricted adversarial examples, the adversary optimizes $z_T$ (obtained via DDIM Inversion) instead

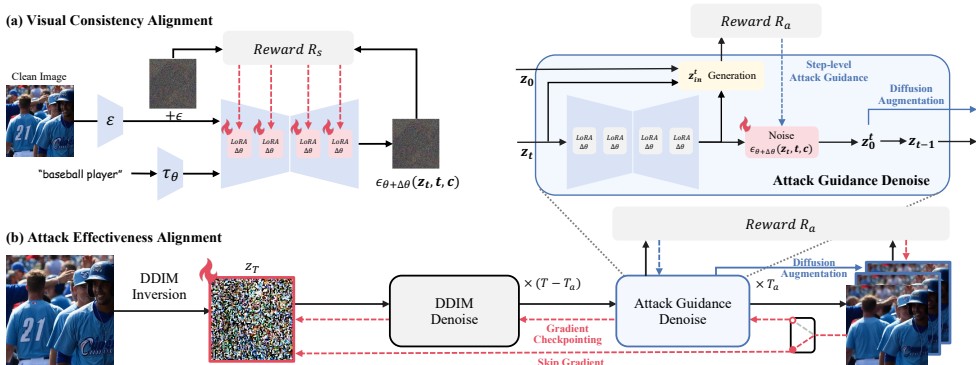

Figure 2: Overview of our APA framework. APA first optimizes the LoRA parameters with a visual consistency reward, storing input image information in LoRA. Then, the input image undergoes DDIM inversion to obtain $z_T$. After DDIM denoising and attack guidance denoising, APA generates trajectory-level $\bar{z}_0$ and diffusion augmentation output $z_0^t$, mixing them using Eq. 12 and passing to the substitute classifier to calculate $R_a$. Finally, $z_T$ is iteratively optimized using skip gradient (APA-SG) or gradient checkpointing (APA-GC).

of directly optimizing $x$ in the pixel space. The mapping from $z_T$ to $x_{adv}$ is defined as $x_{\text{adv}} = \mathcal{D}(\bar{z}_0)$, where $\bar{z}_0$ is obtained by applying $T$ steps of DDIM denoising: $\bar{z}_0 = \underbrace{De \circ \cdots \circ De}_{T}(z_T)$. This

sequential process defines a denoising trajectory. However, these methods attempt to solve Eq. 1 by jointly optimizing $z_T$. Due to the sensitivity of the latent space to noise (shown in Figure 3) and the mutual exclusivity of the two optimization objectives (shown in Figure 5(b)), the generated adversarial examples often fall into a suboptimal trade-off between the conflicting objectives.

To address this, we propose a two-stage adversary preferences alignment framework (APA): first, we reframe unrestricted adversarial attacks as a multi-preference alignment problem and decouple visual consistency and attack effectiveness to independent reward models. Then we strengthen visual consistency via LoRA fine-tuning in the first stage and focus on attack effectiveness through dual-path attack guidance and diffusion augmentation in the second stage. Our APA separates visual consistency and attack effectiveness by independently modeling and aligning preference rewards. It then maximizes attack performance within the optimal solution space of visual consistency, achieving closer Pareto optimality. Figure 2 and Alogrithim 1 present the overall framework of APA.

## 4.1 Visual Consistency Alignment

Diffusion models derive their capabilities from training on extensive images [60, 51], meaning that even minor changes to the latent or prompt can result in substantially different generations. As shown in Figure 3, without perturbations to the $z_T$ obtained via DDIM Inversion, the model nearly reconstructs the clean image after $T$ denoising steps. However, minor noise, particularly adversarial noise, can cause the generated image to lose visual consistency with the original. To preserve visual consistency during adversarial optimization, we aim to strengthen the diffusion model's retention of the input image $x$. A straightforward approach is to fine-tune the

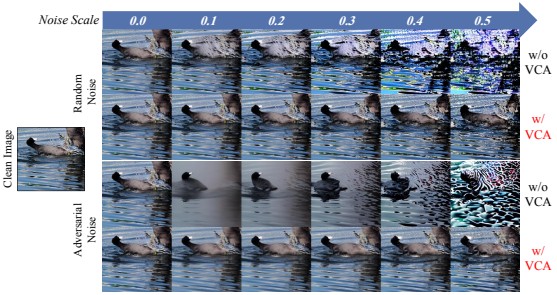

Figure 3: Impact of adversarial and random noise on $z_T$ in generated images. VCA denotes visual consistency alignment. Our LoRA-based VCA, demonstrates improved noise robustness.

UNet to overfit the input image, but this risks catastrophic forgetting, degrades image quality, and is inefficient. Previous research in customized generation [24, 81, 73, 74] suggests that LoRA [29] efficiently encodes high-dimensional image semantics into the low-rank parameter space. Thus, we adopt LoRA $\Delta\theta$ as the policy model during the visual consistency alignment stage. Then, we need to

determine a reward model or reward function $R_s(\cdot)$ to optimize $\Delta\theta$. The straightforward approach is to compute the visual similarity $S(\cdot)$ between the original input and the output of the diffusion model, as follows:

$$\max_{\Delta\theta} R_s(\Delta\theta) = S(\mathcal{D}(\bar{z}_0), x), \tag{5}$$

where $\bar{z}_0$ represents the $T$-step denoised output, requiring $T$ computations of Eq. 3. To reduce this, we first shift the similarity metric to the latent space, calculating $S(\bar{z}_0, z_0)$. We then approximate trajectory-level similarity by accumulating similarity across all steps. Thus, inspired by Eq. 2, $R_s(\Delta\theta)$ is reformulated as:

$$R_s(\Delta\theta) = E_{t\sim[1,T],\epsilon\sim\mathcal{N}(0,\mathbf{I})} - \|\epsilon - \epsilon_{\theta+\Delta\theta}(z_t, t, c)\|^2, \tag{6}$$

Since Eq. 6 is differentiable, we can update $\Delta\theta$ via the direct backpropagation [12, 57] to maximize the reward, as follows: $\Delta\theta = \Delta\theta + \alpha\nabla_{\Delta\theta}R_s$, where $\alpha$ represents the learning rate. Finally, $\Delta\theta$ is integrated into $\epsilon_\theta$, enabling the model to generate visually consistent outputs whether regular noise or adversarial noise is applied to $z_T$, as illustrated in Figure 3.

## 4.2 Attack Effectiveness Alignment

In this stage, We use $z_T$ obtained via DDIM inversion as the optimization variable (optional prompt embedding discussed in Section 5.5). Next, we need to model the reward $R_a$ for attack effectiveness alignment. First, using $f'_\phi(\cdot)$ in Eq. 1 directly as the reward model results in sparse rewards of only 1 or 0, significantly increasing optimization difficulty. To address this, we draw inspiration from traditional transfer attacks [17, 16] and choose a differentiable surrogate model $f_\phi(\cdot)$ as the reward model. This allows optimization via direct backpropagation based on gradients [12, 57] Additionally, to mitigate the gap between the surrogate model and the target model, we propose diffusion augmentation to enhance generalization and alleviate potential reward hacking [22]. Thus, the attack effectiveness reward is formulated as: $R_a(z_T) = L(f_\phi(x_{adv}), y)$, $L(\cdot)$ denotes cross-entropy loss.

**Dual-path Attack Guidance.** We refer to the generation of $x_{adv}$ through $T$-step denoising from $z_T$ as the generation trajectory. Additionally, to enhance gradient consistency, following ACA [10], we use a momentum-based gradient update to optimize $z_T$, encouraging the model to higher $R_a$ output. Thus, our trajectory-level attack optimization can be expressed as:

$$g_{tr} = \nabla_{z_T} R_a(f_\phi(x_{adv}), y), \quad m_{tr}^i = m_{tr}^{i-1} + \frac{g_{tr}}{\|g_{tr}\|_1}, \ z_T = \Pi_{z_T^0+\epsilon_a}(z_T + \mu\cdot\mathrm{sgn}(m_{tr}^i)), \tag{7}$$

where $g_{tr}$ denotes the trajectory-level gradient, $m_{tr}^i$ denotes the momentum of $i^{th}$ trajectory-level attack iteration, $\Pi_{z_T^0+\epsilon_a}$ keeps $z_T$ remain within the $\epsilon_a$-ball centered at the original latent $z_T^0$, $\mathrm{sgn}(\cdot)$ denotes sign function.

Since solving $g_{tr}$ requires computing the gradient across the entire trajectory, direct calculation would require extensive memory. One solution is skip gradient [10], which approximates $g_{tr}$ as $\rho\cdot\nabla_{\bar{z}_0}R_a(f_\phi(x_{adv}), y)$ [10], avoiding memory use for $T$-step denoising. The other is gradient checkpointing [8], which reduces memory during backpropagation by selectively storing intermediate activations, enabling direct computation of $g_{tr}$.

Both skip gradient and gradient checkpointing focus on optimizing $z_T$ from a global trajectory level, where each denoising step uses the same $R_a$. However, different steps contribute uniquely to the final output: larger timesteps affect structure, while smaller ones refine details [43]. As a result, using the same $R_a$ for attack guidance may cause misalignment, reducing attack effectiveness.

To address this issue, we incorporate step-level attack guidance into the denoising steps. Motivated by class-guided generation [15], we introduce the attack reward $R_a$ during each denoising step to guide the noise optimization:

$$\epsilon_{\theta+\Delta\theta}(z_t, t, c) = \epsilon_{\theta+\Delta\theta}(z_t, t, c) - \sqrt{1-\bar{\alpha}_t}\nabla_{z_t}R_a(f_\phi(\mathcal{D}(z_t)), y). \tag{8}$$

Each denoising step adjusts the generation direction based on the current step's $R_a(f_\phi(\mathcal{D}(z_t)), y)$, gradually aligning the final image with higher $R_a$.

Since $z_t$ is an intermediate denoising result, directly inputting it to the classifier biases reward calculation, as classifiers are typically trained on clean samples. A noise-robust classifier could reduce this bias [15], but it would raise training costs and may introduce inconsistencies between

substitute and target classifiers, affecting attack performance. To address this, we first replace $z_t$ with the intermediate result $z_0^t$ generated by DDIM, which represents the estimated trajectory-level $\bar{z}_0$ based on the current step:

$$z_0^t = \frac{z_t - \sqrt{1 - \bar{\alpha}_t}\epsilon_{\theta + \Delta\theta}(z_t, t, c)}{\sqrt{\bar{\alpha}_t}}. \tag{9}$$

Furthermore, given that $z_0^t$ has a bias that increases with larger $t$, we further refine $z_0^t$ by interpolating between the original image's latent $z_0$ and the predicted $z_0^t$, as follows:

$$z_{in}^t = \sqrt{1 - \bar{\alpha}_t}z_0 + \left(1 - \sqrt{1 - \bar{\alpha}_t}\right)z_0^t, \tag{10}$$

where $\sqrt{1 - \bar{\alpha}_t}$ decreases as $t$ decreases, allowing $z_0$ to take on a higher weight at larger $t$ values, making the sample input $x_{in}^t = \mathcal{D}(z_{in}^t)$ to the classifier progressively cleaner, enhancing reward accuracy at each denoising step. Additionally, inspired by the trajectory-level momentum update method, we propose a step-level momentum accumulation:

$$g_{st} = \nabla_{z_t}R_a(f_\phi(x_{in}^t), y), m_{st}^t = m_{st}^{t+1} + \frac{g_{st}}{\|g_{st}\|_1}, \tag{11}$$

where $g_{st}$ and $m_{st}^t$ denotes step-level gradient and momentum. We replace $\nabla_{z_t}R_a(f_\phi(\mathcal{D}(z_t)), y)$ in Eq. 8 with $\text{sgn}(m_{st}^t)$. Finally, by combining trajectory-level and step-level dual-path attack guidance, the generated images are fully aligned with attack effectiveness preference.

**Diffusion Augmentation.** Our dual-path attack optimization is based on direct backpropagation with a differentiable reward. Studies on HPA [47] have shown that direct backpropagation often leads to the diffusion model over-optimizing for the reward model. Similarly, in APA, this causes overfitting to the substitute classifier, limiting transfer attack performance. To address this, we propose diffusion augmentation which uses step-level outputs as data augmentation to enhance the generalization of the trajectory-level gradient $g_{tr}$. Specifically, we collect the step-level $z_0^t$ generated during the denoising using Eq. 9, and mix them with the trajectory-level final output $\bar{z}_0$:

$$x_0^t = \varrho((\mathcal{D}(z_0^t) + \mathcal{D}(\bar{z}_0))/2), \tag{12}$$

where $\varrho(\cdot)$ denotes differentiable data augmentation including random padding, resizing, and brightness adjustment. Appendix F.3 further shows that stronger data transformations (e.g., [72] used in $L_p$ attacks) can further boost performance, underscoring the scalability of our method. Finally, the trajectory-level gradient $g_{tr}$ in Eq. 7 is enhanced to $g_{tr} = \nabla_{z_T}\frac{1}{T}\sum_{t=0}^{T}R_a(f_\phi(x_0^t), y)$.

We collectively refer to step-level attack guidance and diffusion augmentation as the attack guidance denoise process, as shown in Figure 2. To balance time efficiency and image quality, we apply attack guidance denoise only in the final $T_a$ steps. Overall, our framework is implemented as a clean and modular two-stage pipeline. The visual consistency alignment is achieved through lightweight LoRA fine-tuning, and after merging the LoRA parameters into the UNet, no extra parameters are introduced in the subsequent attack alignment stage. The attack alignment is seamlessly integrated into the denoising process by augmenting it with attack guidance. Moreover, our APA framework is highly flexible, allowing for different gradient propagation strategies (APA-SG for skip gradient and APA-GC for gradient checkpointing) and different optimization params (APA-GC-P for text optimization). More details are provided below.

## 5 Experiments

### 5.1 Experimental Settings

**Datasets and Models.** We choose the widely used ImageNet-compatible Dataset [34], consisting of 1,000 images from ImageNet's validation set [14]. Following [10], we select 6 convolutional neural networks (CNNs) and 4 vision transformers (ViTs) as target models for the attack.

**Attack Methods.** We compare with unrestricted attacks including SAE [28], cAdv [2], tAdv [2], ColorFool [63], and NCF [80], as well as diffusion-based methods ACA [10], DiffAttack [7] and DiffPGD [79] in Table 1. Following [10], we use attack success rate (ASR, %), the percentage of misclassified images—as the evaluation metric, reporting both white-box and black-box ASR. **Additional comparisons under different settings are presented in the Appendix**: 1) AdvDiffuser [9] and AdvDiff [13] in Appendix E.3; 2) $L_p$ attacks [16, 79, 17, 45] in Appendix E.2.

Table 1: Attack performance comparison on normally trained CNNs and ViTs. We report attack success rates ASR (%) of each method ("*" means white-box ASR), Avg. ASR refers to the average attack success rate on non-substitute models (black-box ASR).

| Substitute Model | Attack | Models | | | | | | | | | | Avg. ASR (%) |
|---|---|---|---|---|---|---|---|---|---|---|---|---|
| | | CNNs | | | | | | Transformers | | | | |
| | | MN-v2 [61] | Inc-v3 [66] | RN-50 [26] | Dense-161 [30] | RN-152 [26] | EF-b7 [67] | MobViT-s [53] | ViT-B [18] | Swin-B [48] | PVT-v2 [71] | |
| - | Clean | 12.1 | 4.8 | 7.0 | 6.3 | 5.6 | 8.7 | 7.8 | 8.9 | 3.5 | 3.6 | 6.83 |
| MobViT-s | SAE | 60.2 | 21.2 | 54.6 | 42.7 | 44.9 | 30.2 | 82.5* | 38.6 | 21.1 | 20.2 | 37.08 |
| | cAdv | 41.9 | 25.4 | 33.2 | 31.2 | 28.2 | 34.7 | 84.3* | 32.6 | 22.7 | 22.0 | 30.21 |
| | tAdv | 33.6 | 18.8 | 22.1 | 18.7 | 18.7 | 15.8 | 97.4* | 15.3 | 11.2 | 13.7 | 18.66 |
| | ColorFool | 47.1 | 12.0 | 40.0 | 28.1 | 30.7 | 19.3 | 81.7* | 24.3 | 9.7 | 10.0 | 24.58 |
| | NCF | 67.7 | 31.2 | 60.3 | 41.8 | 52.2 | 32.2 | 74.5* | 39.1 | 20.8 | 23.1 | 40.93 |
| | DiffPGD-MI | 59.9 | 42.4 | 48.1 | 44.6 | 38.5 | 38.1 | 95.7* | 29.0 | 24.9 | 42.5 | 40.89 |
| | ACA | 66.2 | 56.6 | 60.6 | 58.1 | 55.9 | 55.5 | 89.8* | 51.4 | 52.7 | 55.1 | 56.90 |
| | DiffAttack | 79.7 | 67.3 | 75.5 | 72.2 | 72.1 | 66.0 | 99.8* | 59.1 | 64.6 | 73.7 | 70.02 |
| | **APA-SG(Ours)** | 81.3 | 66.3 | 73.8 | 71.5 | 68.9 | 65.0 | 98.1* | 51.6 | 46.1 | 68.2 | 65.85 |
| | **APA-GC(Ours)** | **88.3** | **77.1** | **86.6** | **81.2** | **81.2** | **78.4** | **99.4*** | **59.3** | **61.9** | **83.4** | **77.48** |
| MN-v2 | SAE | 90.8* | 22.5 | 53.2 | 38.0 | 41.9 | 26.9 | 44.6 | 33.6 | 16.8 | 18.3 | 32.87 |
| | cAdv | 96.6* | 26.8 | 39.6 | 33.9 | 29.9 | 32.7 | 41.9 | 33.1 | 20.6 | 19.7 | 30.91 |
| | tAdv | 99.9* | 27.2 | 31.5 | 24.3 | 24.5 | 22.4 | 40.5 | 16.1 | 15.9 | 15.1 | 24.17 |
| | ColorFool | 93.3* | 9.5 | 25.7 | 15.3 | 15.4 | 13.4 | 15.7 | 14.2 | 5.9 | 6.4 | 13.50 |
| | NCF | 93.2* | 33.6 | 65.9 | 43.5 | 56.3 | 33.0 | 52.6 | 35.8 | 21.2 | 20.6 | 40.28 |
| | DiffPGD-MI | 97.4* | 54.1 | 68.2 | 57.8 | 56.6 | 52.1 | 68.0 | 28.7 | 22.9 | 41.8 | 50.02 |
| | ACA | 93.1* | 56.8 | 62.6 | 55.7 | 56.0 | 51.0 | 59.6 | 48.7 | 48.6 | 50.4 | 54.38 |
| | DiffAttack | 98.5* | 61.5 | 75.1 | 65.4 | 65.7 | 59.5 | 70.9 | 41.5 | 37.7 | 54.2 | 59.05 |
| | **APA-SG(Ours)** | 99.8* | 80.4 | 88.1 | 83.0 | 81.7 | 78.8 | 78.5 | 55.9 | 39.5 | 63.4 | 72.14 |
| | **APA-GC(Ours)** | **100*** | **91.4** | **97.7** | **95.5** | **95.0** | **91.8** | **93.2** | **74.3** | **59.0** | **85.2** | **87.01** |
| RN-50 | SAE | 63.2 | 25.9 | 88.0* | 41.9 | 46.5 | 28.8 | 45.9 | 35.3 | 20.3 | 19.6 | 36.38 |
| | cAdv | 44.2 | 25.3 | 97.2* | 36.8 | 37.0 | 34.9 | 40.1 | 30.6 | 19.3 | 20.2 | 32.04 |
| | tAdv | 43.4 | 27.0 | 99.0* | 28.8 | 30.2 | 21.6 | 35.9 | 16.5 | 15.2 | 15.1 | 25.97 |
| | ColorFool | 41.6 | 9.8 | 90.1* | 18.6 | 21.0 | 15.4 | 20.4 | 15.4 | 5.9 | 6.8 | 17.21 |
| | NCF | 71.2 | 33.6 | 91.4* | 48.5 | 60.5 | 32.4 | 52.6 | 36.8 | 19.8 | 21.7 | 41.90 |
| | DiffPGD-MI | 75.2 | 60.6 | 96.8* | 75.0 | 78.9 | 55.3 | 67.5 | 30.3 | 26.5 | 48.5 | 57.53 |
| | ACA | 69.3 | 61.6 | 88.3* | 61.9 | 61.7 | 60.3 | 62.6 | 52.9 | 51.9 | 53.2 | 59.49 |
| | DiffAttack | 78.5 | 65.8 | 97.2* | 78.9 | 83.2 | 61.3 | 69.5 | 45.3 | 42.8 | 60.6 | 65.10 |
| | **APA-SG(Ours)** | 89.0 | 83.4 | 99.6* | 89.6 | 90.1 | 77.3 | 76.7 | 58.5 | 45.7 | 67.6 | 75.32 |
| | **APA-GC(Ours)** | **97.6** | **93.5** | **99.7*** | **97.6** | **98.4** | **91.1** | **90.9** | **75.6** | **63.8** | **83.7** | **88.02** |
| ViT-B | SAE | 54.5 | 26.9 | 49.7 | 38.4 | 41.4 | 30.4 | 46.1 | 78.4* | 19.9 | 18.1 | 36.16 |
| | cAdv | 31.4 | 27.0 | 26.1 | 22.5 | 19.9 | 26.1 | 32.9 | 96.5* | 18.4 | 16.9 | 24.58 |
| | tAdv | 39.5 | 22.8 | 25.8 | 23.2 | 22.3 | 20.8 | 34.1 | 93.5* | 16.3 | 15.3 | 24.46 |
| | ColorFool | 45.3 | 13.9 | 35.7 | 24.3 | 28.8 | 19.8 | 27.0 | 83.1* | 8.9 | 9.3 | 23.67 |
| | NCF | 55.9 | 25.3 | 50.6 | 34.8 | 42.3 | 29.9 | 40.6 | 81.0* | 20.0 | 19.1 | 35.39 |
| | DiffPGD-MI | 59.5 | 40.9 | 44.2 | 41.9 | 41.3 | 41.3 | 52.2 | 95.4* | 33.8 | 44.1 | 44.13 |
| | ACA | 64.6 | 58.8 | 60.2 | 58.1 | 58.1 | 57.1 | 60.8 | 87.7* | 55.5 | 54.9 | 58.68 |
| | DiffAttack | 47.2 | 44.2 | 44.3 | 42.9 | 44.5 | 44.8 | 49.6 | 94.5* | 46.8 | 41.3 | 45.06 |
| | **APA-SG(Ours)** | 69.3 | 67.6 | 67.5 | 66.8 | 65.4 | 70.0 | 67.6 | 99.2* | 62.5 | 59.2 | 66.21 |
| | **APA-GC(Ours)** | **77.0** | **74.8** | **75.4** | **75.9** | **75.4** | **76.8** | **74.5** | **98.4*** | **73.4** | **70.2** | **74.82** |

**Implementation Details.** We set attack guidance step $T_a = 10$, attack iterations $N = 10$, attack scale $\epsilon_a = 0.4$, and attack step size $\mu = 0.04$. APA-SG adopts the entire inversion step of $T = 50$. APA-GC adopts $T = 10$ to improve efficiency. Our work is based on Stable Diffusion V1.5 [60]. During the visual consistency alignment phase, we fine-tune only the projection matrices Q, K, and V in the attention modules of the UNet with each clean image. With the LoRA rank set to 8, we train for 200 steps. Experimental results indicate that APA-GC delivers strong attack performance. Thus, we add visual consistency constraints to APA-GC's $R_a$ without concerns about impacting attack performance, setting $R_a = R_a - \lambda \|z_0 - \bar{z}_0\|_2$ with $\lambda = 10$.

## 5.2 Attack Performance Comparison

To evaluate the performance of our APA framework, we select 10 models as target models, including both CNN and transformer architectures. The attack performance comparison is shown in Table 1.

**Non-diffusion-based Attacks.** Texture-based tAdv achieves a higher white-box ASR but lower black-box ASR than color-based methods such as NCF and SAE. NCF demonstrates the highest transferability, achieving an average ASR of 39.6% across four models. Our APA, which modifies multiple input semantics simultaneously, surpasses single-semantic attacks, with notable improvements and 30.2% (APA-SG) and 42.2% (APA-GC) in black-box ASR across four models than NCF.

**Diffusion-based Attacks.** DiffPGD-MI generates unrestricted adversarial examples by first applying $L_p$ norm perturbations, and then performing diffusion-based image translation. ACA and DiffAttack

Table 2: Attack performance on adversarial defense methods, ViT-B as the substitute model.

| Attack | HGD | R&P | NIPS-r3 | JPEG | Bit-Red | DiffPure | Inc-v3$_{ens3}$ | Inc-v3$_{ens4}$ | IncRes-v2$_{ens}$ | Res-De | Shape-Res | ViT-B-CvSt | Avg. ASR (%) |
|---|---|---|---|---|---|---|---|---|---|---|---|---|---|
| Clean | 1.2 | 1.8 | 3.2 | 6.2 | 17.6 | 15.4 | 6.8 | 8.9 | 2.6 | 4.1 | 6.7 | 8.4 | 6.91 |
| SAE | 21.4 | 19.0 | 25.2 | 25.7 | 43.5 | 39.8 | 25.7 | 29.6 | 20.0 | 35.1 | 49.6 | 38.9 | 31.13 |
| cAdv | 12.2 | 14.0 | 17.7 | 11.1 | 33.9 | 32.9 | 19.9 | 23.2 | 14.6 | 16.2 | 25.3 | 20.6 | 20.13 |
| tAdv | 10.9 | 12.4 | 14.4 | 17.8 | 29.6 | 21.2 | 17.7 | 19.0 | 12.5 | 16.4 | 25.4 | 11.3 | 17.38 |
| ColorFool | 9.1 | 9.6 | 15.3 | 18.0 | 37.9 | 33.8 | 17.8 | 21.3 | 10.5 | 20.3 | 35.0 | 31.2 | 21.65 |
| NCF | 22.8 | 21.1 | 25.8 | 26.8 | 43.9 | 39.6 | 27.4 | 31.9 | 21.8 | 34.4 | 47.5 | 35.8 | 31.57 |
| DiffPGD-MI | 25.7 | 28.3 | 29.9 | 32.2 | 38.8 | 27.4 | 32.1 | 32.9 | 28.1 | 40.0 | 45.5 | 19.7 | 31.72 |
| ACA | 52.2 | 53.6 | 53.9 | 59.7 | 63.4 | 63.7 | 59.8 | 62.2 | 53.6 | 55.6 | 60.8 | **51.1** | 57.47 |
| DiffAttack | 33.3 | 34.3 | 33.2 | 37.9 | 47.0 | 38.3 | 38.0 | 42.6 | 35.7 | 40.6 | 45.5 | 19.7 | 37.17 |
| **APA-SG(Ours)** | 61.5 | 61.0 | 63.8 | 66.7 | 71.0 | 63.0 | 66.8 | 67.4 | 63.1 | 64.5 | 68.7 | 45.6 | 63.59 |
| **APA-GC(Ours)** | **73.5** | **71.1** | **72.4** | **72.4** | **74.3** | **71.2** | **72.5** | **73.6** | **71.4** | **72.4** | **75.5** | 42.1 | **70.20** |

directly optimize the input image's latent space, generally achieving higher black-box transferability, especially across architectures. Our method incorporates dual-path attack guidance and diffusion augmentation, enabling APA-SG (with the same gradient backpropagation as ACA) to improve black-box ASR by 12.5%, 10.0% while APA-GC improves black-box performance by 24.4%, 21.9% over ACA and DiffAttack across four models.

**Overall.** Our method outperforms existing unrestricted adversarial attacks in terms of black-box transferability, whether within CNN or transformer architectures or in cross-architecture attacks. Additionally, due to its more precise gradient-guided attack, APA-GC achieves an average performance improvement of 11.9% over APA-SG.

## 5.3 Attacks on Adversarial Defense

To evaluate unrestricted adversarial attacks against existing defenses, we select adversarially trained models (Inc-v3$ens3$, Inc-v3$ens4$, Inc-v2$ens$[68], ViT-B-CvSt[64]) and preprocessing defenses (HGD [44], R&P [76], NIPS-r3, JPEG [25], Bit-Red [78], DiffPure [55]). Additionally, shape-texture debiased models (ResNet50-Debiased (Res-De)[42], Shape-ResNet (Shape-Res)[23]) are selected to counter unrestricted adversarial examples, as shown in Table 2. We use ViT-B as the substitute model and Inc-v3$ens3$ as the target model for input preprocessing defenses. Since existing defenses mainly address $L_p$ attacks, they remain ineffective against unrestricted adversarial attacks. With an advanced APA framework, our APA-GC achieves a 12.7% improvement on Avg. ASR over SOTA method.

## 5.4 Visual Quality Comparison

We compare the visual performance of the top six attack methods based on attack performance in Table 1, using RN-50 as the substitute model.

**Quantitative Comparison.** We use reference-based metrics (LPIPS, SSIM, CLIP Score [59]) to evaluate visual similarity in terms of distribution, structure, and semantics, and no-reference aesthetic metrics (NIMA-AVA [54], CNN-IQA [5]) to assess aesthetic quality, as shown in Table 3. 1) Our visual consistency alignment (VCA) maintains strong visual consistency while achiev-

Table 3: Quantitative comparison of image quality. VCA denotes only using visual consistency alignment. APA-GC-P denotes prompt-based optimization.

| Attack | LPIPS↓ | SSIM↑ | CLIP Score↑ | NIMA-AVA↑ | CNN-IQA↑ | Avg. ASR↑ |
|---|---|---|---|---|---|---|
| Clean | 0.00 | 1.00 | 1.00 | 4.99 | 0.58 | 6.83 |
| DDIM Inversion | 0.19 | 0.73 | 0.89 | 5.03 | 0.63 | 22.10 |
| VCA | 0.05 | 0.85 | 0.97 | 5.13 | 0.63 | 7.60 |
| SAE | 0.43 | 0.79 | 0.81 | 5.00 | 0.53 | 36.38 |
| cAdv | 0.15 | **0.98** | 0.88 | 4.85 | 0.55 | 32.04 |
| NCF | 0.40 | 0.83 | 0.83 | 4.97 | 0.57 | 41.90 |
| DiffPGD-MI | 0.29 | 0.71 | 0.85 | 4.69 | 0.61 | 57.53 |
| ACA | 0.37 | 0.61 | 0.79 | 5.38 | 0.65 | 59.49 |
| DiffAttack | 0.14 | 0.68 | 0.87 | 5.17 | 0.66 | 65.10 |
| **APA-SG(Ours)** | 0.25 | 0.67 | 0.86 | 5.29 | 0.62 | 75.32 |
| **APA-GC(Ours)** | 0.23 | 0.69 | 0.83 | **5.39** | **0.67** | **88.02** |
| **APA-GC-P(Ours)** | **0.09** | 0.82 | **0.91** | 5.22 | 0.63 | 62.08 |

ing a higher aesthetic score compared to clean images and original DDIM Inversion. 2) Our method achieves higher aesthetic scores compared to non-diffusion-based attacks and DiffPGD owing to stable diffusion's strong generative capabilities. 3) Our APA achieves comparable visual similarity and aesthetic scores to ACA and DiffAttack. 4) Within our framework, APA-GC-P which optimizes prompt instead of latent (see Section 5.5) has the best visual consistency.

**Qualitative Comparison.** Due to the inability of quantitative metrics to fully measure visual consistency, we perform a qualitative analysis in Figure 4. SAE with NCF alters the original style, DiffPGD-MI introduces noticeable perturbations, and cAdv affects authenticity by changing colors.

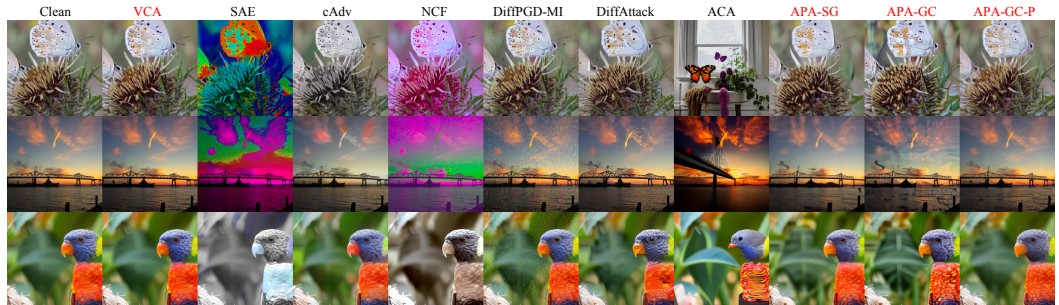

Figure 4: Qualitative comparison of image quality.

ACA disrupts the original structure, while DiffAttack makes certain parts of the main subject appear overly sharp and unnatural. In contrast, our APA preserves structure and color, making only subtle, natural adjustments mainly in the background, resulting in a more visually consistent effect.

## 5.5 Ablation Studies

The previous section has discussed the ablation of visual consistency alignment (Figure 3) and gradient backpropagation (Table 1). Here, we focus on analyzing the remaining key modules and design. All experiments utilize RN-50 as the substitute model. Time analysis is discussed in Appendix A.

**Key Modules.** Rows 1 and 2 of Table 4 show that the dual-path attack guidance module improves black-box attack performance by 6.6% compared to only trajectory-level guidance. To further validate the superiority of our attack guidance, we re-implement class-guided [15] and Upainting [40] by ap-

Table 4: Ablation studies on key modules. L denotes latent-based optimization, P denotes prompt-based optimization.

| Optimized Params | Dual-path Guidance | Diffusion Augmentation | Backpro-pagation | White-box ASR(%) | Black-box ASR(%) |
|---|---|---|---|---|---|
| L | | | SG | 96.8 | 48.28 |
| L | ✓ | | SG | **99.7** | 54.88 |
| L | | ✓ | SG | 92.1 | 62.38 |
| L | ✓ | ✓ | SG | 99.6 | 75.32 |
| L | ✓ | ✓ | GC | **99.7** | **88.02** |
| P | ✓ | ✓ | GC | 99.5 | 62.08 |

plying $R_a$ for adversary preferences alignment. Figure 5(a) shows improved attack performance with our method, which benefits from more accurate attack reward guidance through clear $z_{in}^t$ in Eq. 10 and step-level momentum accumulation in Eq. 11. Rows 1 and 3 of Table 4 demonstrate that diffusion augmentation mitigates the limitations of direct backpropagation overfitting to the substitute model, improving black-box performance by 14.1% with only a slight decrease in white-box performance. Rows 3 and 4 in Table 4 show that diffusion augmentation combined with dual-path attack guidance effectively improves black-box attack performance.

**Two-stage vs. One-stage Alignment.** To validate the advantages of our two-stage alignment, we adapt APA into a single-stage alignment (APA*): replacing LoRA-based visual alignment and incorporating joint optimization in the second stage, i.e., $R_a = R_a - \lambda\|z_0 - \bar{z}_0\|_2$. Experimental results in Figure 5(b) demonstrate: 1) One-stage alignment (both APA* and ACA) suffer from reward hacking due to conflicting objectives during joint optimization (as $\lambda$ increases, Avg. ASR decreases while SSIM increases). 2) Our two-stage APA maximizes attack performance within the optimal solution space of visual consistency, achieving closer Pareto optimality.

**Optimized Parameters.** Row 6 in Table 4 shows the attack performance with prompt-based optimization (APA-GC-P), which optimizes text features $\tau_\theta(c)$ with gradient checkpointing. Compared to the direct correspondence between the latent and image spaces, prompt-based optimization indirectly guides image generation through $\epsilon_{\theta+\Delta\theta}(z_t, t, c)$, resulting in lower attack efficacy than APA-GC. However, as shown in Table 3, APA-GC-P demonstrates improved visual consistency, offering attackers greater flexibility depending on the application scenario.

**Scalability.** To demonstrate the flexibility and scalability of our framework, we extend it to various diffusion models (e.g., ControlNet [39]) in Appendix F.1 and different tasks including targeted attacks, visual question answering, and object detection in Appendix F.2.

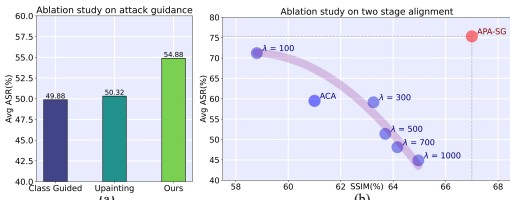
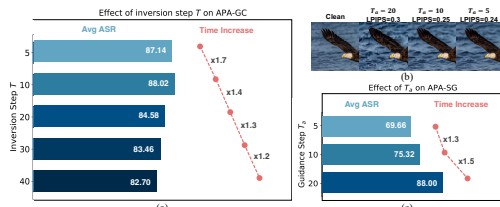

Figure 5: (a) Comparison with different optimization guidance. (b) Comparison of our two-stage APA-SG and one-stage alignment under $\lambda$-controlled visual consistency.

Figure 6: (a) denotes hyper-parameters tuning on $T$. (b) and (C) denotes hyper-parameters tuning on $T_a$. RN-50 as the substitute model.

## 5.6 Hyper-parameters tuning

**Guidance Step** $T_a$. Figure 6 (b), (c) show the impact of $T_a$ on performance. As $T_a$ increases, attack performance improves, time cost rises, and image quality deteriorates. Considering these factors, we choose $T_a = 10$.

**Inversion Step** $T$. APA-GC employs gradient checkpointing to save memory at the cost of additional time. To improve efficiency, we investigate the impact of reducing inversion steps $T$ on performance. Figure 6(a) shows that setting $T$ below $T_a$ reduces attack performance due to insufficient guidance, while exceeding $T_a$ also degrades attack performance due to bias introduced by overly deep gradient chains. Thus, we set $T = T_a = 10$.

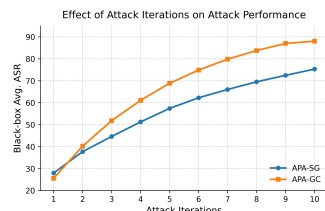

Figure 7: Effect of attack iterations on attack performance.

**Attack Iterations** $N$. We analyze the effect of the number of attack iterations on performance, as shown in Figure 7. The attack performance of our APA improves rapidly with increasing iterations, where APA-GC, benefiting from more accurate gradient computation, exhibits faster improvement and earlier convergence. Together with Table 1, we observe that APA-SG surpasses ACA within only six iterations, while APA-GC achieves this in just four.

## 6 Conclusion

In this paper, we broaden the application of preference alignment, reformulating unrestricted adversarial example generation as an adversary preferences alignment problem. However, the inherently conflicting objectives of visual consistency and attack effectiveness significantly increase the difficulty of alignment. To address this challenge, we propose Adversary Preferences Alignment (APA), a two-stage framework that first establishes visual consistency through LoRA-based alignment guided by a rule-based similarity reward, and then enhances attack effectiveness via dual-path attack guidance and diffusion-based augmentation. Experimental results demonstrate that APA achieves superior black-box transferability while preserving high visual consistency. We hope our work serves as a bridge between preference alignment and adversarial attacks, and inspires further research on adversarial robustness from an alignment perspective.

**Broader Impacts.** Our work is centered around unrestricted adversarial examples, aiming to deepen the understanding of model vulnerability and enhance the robustness and reliability of deep learning models. The proposed methodology and research results are intended to be used only for academic and ethical purposes, such as enhancing security protection capabilities as well as facilitating the safe implementation of AI technologies.

**Ackowledgements.** This work was supported by National Natural Science Foundation of China (No.62576109, 62072112).

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

## Appendix

This appendix is structured as follows:

- In Appendix A, we provide time analysis of our APA.

- In Appendix B, we provide limitations.

- In Appendix C, we provide pseudo code of our APA.

- In Appendix D, we provide more diagnostic experiments.

- In Appendix E, we provide more attack methods compared with our method, including unrestricted attacks (ADef [1], ACE [83], ReColorAdv [35], PPGD [36], LPA [36], AdvDiff [13]) and $L_p$ attacks (MI [16], NI [45], DIM [79], TIM [17]).

- In Appendix F, we conduct extensive experiments to evaluate our method across various diffusion models (e.g., ControlNet [39]) and downstream tasks, including targeted attacks, visual question answering, and object detection. Moreover, we further show that stronger data transformations (e.g., [70, 72] used in $L_p$ attacks) can further boost performance, underscoring the scalability of our method.

- In Appendix G, we provide more visualizations.

## A  Time Analysis

Most unrestricted attacks, as well as our method, fall under the image-specific framework. To further demonstrate the advantages of our proposed APA framework, we conduct a comprehensive time efficiency analysis. Our empirical evaluation on an NVIDIA A100 GPU shows that visual consistency alignment requires 38 seconds, while attack effectiveness alignment (APA-SG) takes 58.5 seconds. When combined, our complete APA framework requires a total execution time of 96.5 seconds. This represents a significant 16% improvement in computational efficiency compared to the ACA method, which requires 114.8 seconds on an NVIDIA A100 GPU.

## B  Discussion & Limitations

Adversarial examples have been a long-standing research hotspot in the field of artificial intelligence. Previous works have explored various directions to design stronger adversarial examples, such as gradient-based optimization [52], heuristic optimization [32],diffusion-based methods [10, 11, 55, 31], and [19], etc. In contrast, our method is the first to approach unrestricted adversarial attacks from a preference alignment perspective. The key idea lies in alignment-driven modeling, where we define and quantify malicious preferences (visual consistency and attack effectiveness), select suitable alignment strategies (e.g., DPO, RL, or direct backpropagation), and build a stable alignment framework (joint or decoupled). A two-stage alignment framework is proposed to decouple inherently conflicting preferences—maximizing attack performance under visual consistency constraints—approaching the Pareto frontier. To further stabilize training, dual-path optimization and diffusion augmentation are introduced to mitigate overfitting in direct backpropagation. Extensive experiments demonstrate the effectiveness, flexibility, and robustness of our approach.

We recognize two primary limitations of our work: 1) While APA achieves greater time efficiency than previous diffusion-based attacks (e.g., ACA), it is still more computationally expensive than conventional $L_p$ attacks. One of the costs arises from DDIM sampling; thus, integrating faster samplers could further enhance efficiency. 2) This work is limited to diffusion models, and does not explore the applicability of our method to other generative frameworks such as rectified flows [46, 20]. We plan to investigate this in future work.

## C  Pseudo Code of APA

We provide pseudo code of our APA framework in Algorithm 1 and Symbol Table in Table 5.

**Algorithm 1:** Our APA Framework

**Input**: input image $x$, prompt $c$, label $y$, substitute classifier $f_\phi$, denoising model $\epsilon_\theta$, encoder $\mathcal{E}$, decoder $\mathcal{D}$, inversion step $T$, attack guidance denoising step $T_a$, $m_{tr} = 0$.

**Output**: unrestricted adversarial example $x_{adv}$

1: $\epsilon_{\theta+\Delta\theta} = $ Visual_Consistency_Alignment$(x, c, \epsilon_\theta)$.
2: $z_0 = \mathcal{E}(x)$.
3: Generate $z_T$ using DDIM Inversion, $z_T^0 = z_T$.
4: **for** $i$ in $[1, N+1]$ **do**
5:      $m_{st} = 0, \mathrm{V} = \{\}$.
6:      **for** $t$ in $[T, 1]$ **do**
7:          **if** $t > T_a$ **then**
8:              Calculate $z_{t-1}$ using DDIM Denoising.
9:          **else**
10:             Calculate $z_{in}^t$ using Eq. 11.
11:             $g_{st} = \nabla_{z_t} R_a(f_\phi(\mathcal{D}(z_{in}^t)), y)$.
12:             $m_{st} = m_{st} + \frac{g_{st}}{\|g_{st}\|_1}$.
13:             $\epsilon_{\theta+\Delta\theta}(z_t, t, c) -= \sqrt{1 - \bar{\alpha}_t} \cdot \mathrm{sgn}(m_{st})$.
14:             $z_0^t = \frac{z_t - \sqrt{1 - \bar{\alpha}_t}\epsilon_{\theta+\Delta\theta}(z_t, t, c)}{\sqrt{\bar{\alpha}_t}}$.
15:             $\mathrm{V} = \mathrm{V} + \{z_0^t\}$
16:             $z_{t-1} = \sqrt{\bar{\alpha}_{t-1}}z_0^t + \sqrt{1 - \bar{\alpha}_{t-1}}\epsilon_{\theta+\Delta\theta}(z_t, t, c)$.
17:          **end if**
18:      **end for**
19:      $r = 0$.
20:      **for** $z_0^t$ in $\mathrm{V}$ **do**
21:          $x_0^t = \varrho((\mathcal{D}(z_0^t) + \mathcal{D}(\bar{z}_0))/2)$.
22:          $r = r + R_a(f_\phi(x_0^t), y)$.
23:      **end for**
24:      **if** Skip Gradient **then**
25:          $g_{tr} = \rho \cdot \nabla_{\bar{z}_0} \frac{1}{T_a} r, m_{tr} = m_{tr} + \frac{g_{tr}}{\|g_{tr}\|_1}$.
26:      **else**
         $g_{tr} = \nabla_{z_T} \frac{1}{T_a} r, m_{tr} = m_{tr} + \frac{g_{tr}}{\|g_{tr}\|_1}$.
27:      **end if**
28:      $z_T = \Pi_{z_T^0 + \epsilon_a}(z_T + \mu \cdot \mathrm{sgn}(m_{tr}))$.
29: **end for**
30: **return** $\mathcal{D}(\bar{z}_0)$

Table 5: Symbol Table

| Symbol | Description |
|---|---|
| $z_T$ | The latent updated at each iteration, initialized as $z_T^0$ |
| $z_T^0$ | Latent representation obtained by applying DDIM inversion to the original image latent for $T$ steps |
| $z_0$ | VAE-encoded latent of the original image |
| $\bar{z}_0$ | Reconstructed $z_0$ obtained by DDIM denoising for $T$ steps |
| $z_0^t$ | Reconstructed latent based on $z_t$, as defined in Eq. 9; $z_0^0 = \bar{z}_0$ |
| $z_{in}^t$ | Interpolation between $z_0$ and $z_0^t$ |
| $g_{tr}$ | Gradient at the trajectory level |
| $m_{tr}$ | Momentum at the trajectory level |
| $g_{st}$ | Gradient at the step level |
| $m_{st}$ | Momentum at the step level |

# D    More Diagnostic Experiments

## D.1   LoRA Rank

Regarding the LoRA insertion strategy, we adhered to the default configuration for LoRA fine-tuning on UNet. Additionally, we investigated the impact of the LoRA rank on VCA performance. Using 50 randomly sampled images (source model: RN-50), we evaluated two aspects: (1) the effect of rank on reconstruction quality in a non-adversarial setting, and (2) the effect of rank on performance under attack (see Table 6). Overall, the results are consistent with intuition: higher ranks improve visual

Table 6: The effect of LoRA rank on APA.

| Rank | w/o Attack | | w/ Attack | |
|------|------------|---------|------------|---------|
| | CLIP Score | Avg ASR | CLIP Score | Avg ASR |
| 0 | 0.88 | 21.1 | 0.62 | 96.2 |
| 4 | 0.97 | 7.2 | 0.87 | 79.2 |
| 8 | 0.97 | 6.0 | 0.89 | 74.5 |
| 16 | 0.97 | 5.0 | 0.91 | 71.2 |

Table 8: More attack performance comparison on normally trained CNNs and ViTs. We report attack success rates ASR (%) of each method ("*" means white-box ASR), Avg. ASR refers to the average attack success rate on non-substitute models (black-box ASR).

| Substitute Model | Attack | Models | | | | | | | | | | Avg. ASR (%) |
|------------------|--------|--------|--------|--------|--------|--------|--------|--------|--------|--------|--------|--------------|
| | | CNNs | | | | | | Transformers | | | | |
| | | MN-v2 | Inc-v3 | RN-50 | Dense-161 | RN-152 | EF-b7 | MobViT-s | ViT-B | Swin-B | PVT-v2 | |
| - | Clean | 12.1 | 4.8 | 7.0 | 6.3 | 5.6 | 8.7 | 7.8 | 8.9 | 3.5 | 3.6 | 6.83 |
| MobViT-s | ADef | 14.5 | 6.6 | 9.0 | 8.0 | 7.1 | 9.8 | 80.8* | 9.7 | 5.1 | 4.6 | 8.27 |
| | ReColorAdv | 37.4 | 14.7 | 26.7 | 22.4 | 21.0 | 20.8 | 96.1* | 21.5 | 16.3 | 16.7 | 21.94 |
| | PPGD | 15.7 | 7.0 | 9.4 | 8.8 | 7.2 | 10.5 | 100.0* | 9.6 | 5.8 | 5.5 | 8.83 |
| | LPA | 29.5 | 15.0 | 18.7 | 17.6 | 15.5 | 17.1 | 100.0* | 12.5 | 14.1 | 17.5 | 17.50 |
| | ACE | 30.7 | 9.7 | 20.3 | 16.3 | 14.4 | 13.8 | 99.2* | 16.5 | 6.8 | 5.8 | 14.92 |
| | **APA-SG(Ours)** | 81.3 | 66.3 | 73.8 | 71.5 | 68.9 | 65.0 | 98.1* | 51.6 | 46.1 | 68.2 | 65.85 |
| | **APA-GC(Ours)** | **88.3** | **77.1** | **86.6** | **81.2** | **81.2** | **78.4** | 99.4* | **59.3** | **61.9** | **83.4** | **77.48** |
| MN-v2 | ADer | 56.6* | 7.6 | 8.4 | 7.7 | 7.1 | 10.9 | 11.7 | 9.5 | 4.5 | 4.5 | 7.99 |
| | ReColorAdv | 97.7* | 18.6 | 33.7 | 24.7 | 26.4 | 20.7 | 31.8 | 17.7 | 12.2 | 12.6 | 22.04 |
| | PPGD | 99.9* | 10.4 | 14.0 | 11.9 | 11.9 | 13.5 | 14.9 | 10.1 | 6.7 | 6.6 | 11.11 |
| | LPA | **100.0*** | 21.2 | 27.5 | 23.1 | 21.4 | 21.9 | 29.3 | 12.2 | 10.6 | 12.6 | 19.98 |
| | ACE | 99.1* | 9.5 | 17.9 | 12.4 | 12.6 | 11.7 | 16.3 | 12.1 | 5.4 | 5.6 | 11.50 |
| | **APA-SG(Ours)** | 99.8* | 80.4 | 88.1 | 83.0 | 81.7 | 78.8 | 78.5 | 55.9 | 39.5 | 63.4 | 72.14 |
| | **APA-GC(Ours)** | **100*** | **91.4** | **97.7** | **95.5** | **95.0** | **91.8** | **93.2** | **74.3** | **59.0** | **85.2** | **87.01** |
| RN-50 | ADer | 15.5 | 7.7 | 55.7* | 8.4 | 7.8 | 11.4 | 12.3 | 9.2 | 4.6 | 4.9 | 9.09 |
| | ReColorAdv | 40.6 | 17.7 | 96.4* | 28.3 | 33.3 | 19.2 | 29.3 | 18.8 | 12.9 | 13.4 | 23.72 |
| | PPGD | 23.1 | 12.3 | 99.7* | 16.6 | 18.0 | 13.3 | 14.9 | 10.6 | 6.3 | 6.9 | 13.56 |
| | LPA | 37.6 | 24.0 | **100.0*** | 34.4 | 38.0 | 22.0 | 29.2 | 13.5 | 12.2 | 14.3 | 25.02 |
| | ACE | 32.8 | 9.4 | 99.1* | 16.1 | 15.2 | 12.7 | 20.5 | 13.1 | 6.1 | 5.3 | 14.58 |
| | **APA-SG(Ours)** | 89.0 | 83.4 | 99.6* | 89.6 | 90.1 | 77.3 | 76.7 | 58.5 | 45.7 | 67.6 | 75.32 |
| | **APA-GC(Ours)** | **97.6** | **93.5** | 99.7* | **97.6** | **98.4** | **91.1** | **90.9** | **75.6** | **63.8** | **83.7** | **88.02** |
| ViT-B | ADer | 15.3 | 8.3 | 9.9 | 8.4 | 7.6 | 12.0 | 12.4 | 81.5* | 5.3 | 5.5 | 9.41 |
| | ReColorAdv | 25.5 | 12.1 | 17.5 | 13.9 | 14.4 | 15.4 | 22.9 | 97.7* | 10.9 | 8.6 | 15.69 |
| | PPGD | 15.9 | 7.5 | 8.9 | 8.3 | 7.9 | 10.3 | 10.9 | 99.7* | 5.6 | 3.9 | 8.80 |
| | LPA | 21.4 | 10.4 | 13.9 | 12.5 | 11.6 | 14.5 | 16.6 | **100.0*** | 9.1 | 7.8 | 13.09 |
| | ACE | 30.9 | 11.4 | 22.0 | 15.5 | 15.2 | 13.0 | 17.0 | 98.6* | 6.5 | 6.3 | 15.31 |
| | **APA-SG(Ours)** | 69.3 | 67.6 | 67.5 | 66.8 | 65.4 | 70.0 | 67.6 | 99.2* | 62.5 | 59.2 | 66.21 |
| | **APA-GC(Ours)** | **77.0** | **74.8** | **75.4** | **75.9** | **75.4** | **76.8** | **74.5** | 98.4* | **73.4** | **70.2** | **74.82** |

fidelity but reduce attack success rates and increase both computational and training costs due to a larger number of tunable parameters. Considering this trade-off, we find that a rank of 8 achieves a favorable balance.

## D.2 Visual Consistency Alignment

Figure 3 qualitatively illustrates the importance of VCA in maintaining visual consistency. The first three rows of Table 3 demonstrate that VCA improves the reconstruction quality of clean samples. We also evaluated the image quality when only attack effectiveness alignment was

Table 7: Performance comparison of APA-SG with and without VCA.

| Method | LPIPS | SSIM | CLIP Score | Avg. ASR |
|--------|-------|------|------------|----------|
| w/o VCA | 0.55 | 0.46 | 0.62 | 95.43 |
| w/ VCA | 0.25 | 0.67 | 0.86 | 75.32 |

applied (i.e., without VCA) in Table 7. The noticeable degradation in visual quality in this setting underscores the essential role of VCA in preserving visual consistency during attack alignment.

.

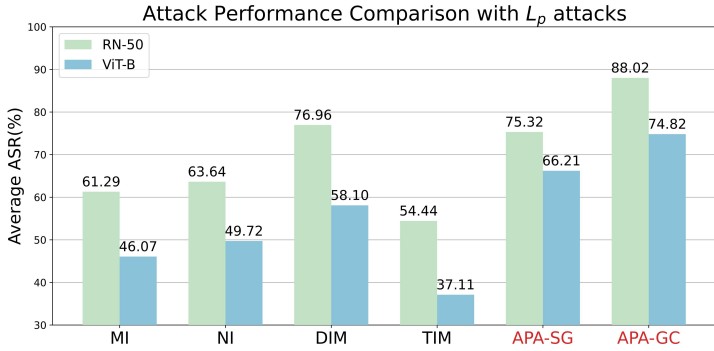

Figure 8: Attack performance comparison with $L_p$ attacks. RN-50 and ViT-B as substitute models. Average ASR refers to the average attack success rate on non-substitute models.

# E  More Attack Performance Comparison

## E.1  More Unrestricted Attacks

As a supplement to Table 1 in the main body, we include a comparison of the attack performance of four additional methods in Table 8: ADef [1], ACE [83], ReColorAdv [35], PPGD [36], and LPA [36], against our APA. Our findings demonstrate that, regardless of whether a CNN or ViT model is used as the substitute model, the transfer attack performance of our method significantly surpasses these four methods.

## E.2  $L_p$ Attacks.

To further validate the robustness of our method, we compare our method with classical transfer attacks with $L_p$ attacks ($L_\infty = 16/255$): gradient optimization attacks (MI [16], NI [45]) and input transformations attacks (DIM [77], and TIM [17]), as shown in Figure 8. Although this comparison is inherently unfair for ours (as unrestricted adversarial examples are more natural), our methods consistently achieve superior black-box transferability, especially when using ViT as the substitute model.

## E.3  AdvDiff

AdvDiff [13] explores a non-traditional form of unrestricted adversarial examples, which cannot specify a reference image and instead generates adversarial examples starting from random noise. (AdvDiffuser [9] adopts a similar approach; however, AdvDiff has demonstrated superior performance compared to AdvDiffuser, and since the code for AdvDiffuser is not publicly available, we include comparisons exclusively with AdvDiff, considering both the AdvDiff and AdvDiff-Untarget versions [13].) To highlight the advantages of our method, we follow the experimental setup of AdvDiff for comparison. Specifically, we first run AdvDiff without adversarial optimization to generate clean, original images. These images are then used as reference images. The results are presented in Table 9. We find that our method demonstrates robust attack performance regardless of whether RN-50 or ViT is used as the substitute model. In contrast, AdvDiff shows significant performance discrepancies between RN-50 and ViT. Overall, our method achieves superior attack performance.

Figure 9 shows the visualizations including our method and AdvDiff. We observe that while AdvDiff-Untarget shows improved attack performance compared to AdvDiff, its image generation quality is significantly lower. In contrast, our method achieves superior performance in both image generation quality and attack effectiveness.

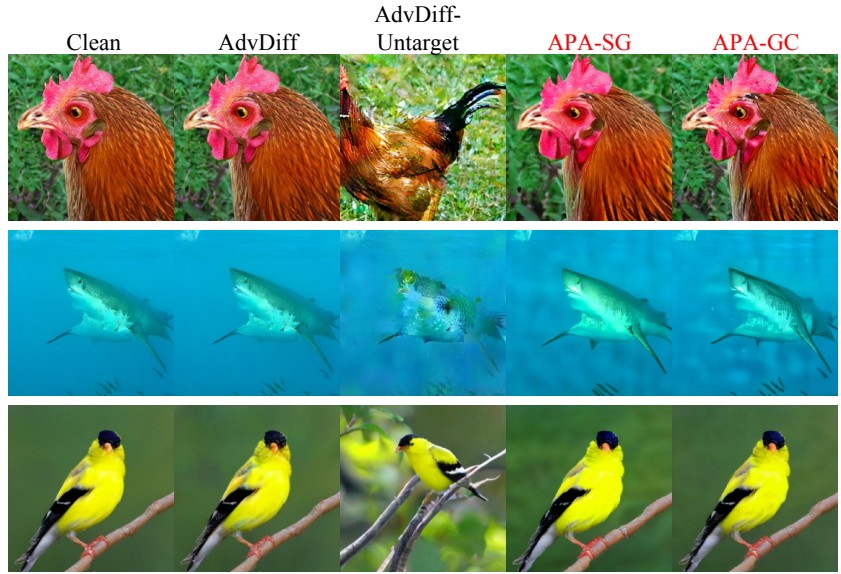

| Clean | AdvDiff | AdvDiff-Untarget | APA-SG | APA-GC |

Figure 9: Visual quality comparison with AdvDiff using RN-50 as the substitute model.

Table 9: Attack performance comparison with AdvDiff and AdvDiff-Untarget.

| Substitute Model | Attack | Models | | | | | | | | | | Avg. ASR (%) |
|---|---|---|---|---|---|---|---|---|---|---|---|---|
| | | CNNs | | | | | | Transformers | | | | |
| | | MN-v2 | Inc-v3 | RN-50 | Dense-161 | RN-152 | EF-b7 | MobViT-s | ViT-B | Swin-B | PVT-v2 | |
| - | Clean | 6.2 | 7.6 | 7.3 | 6.5 | 6.5 | 7.2 | 7.4 | 6.2 | 5.7 | 4.8 | 6.54 |
| RN-50 | AdvDiff | 9.3 | 9.9 | **100.0*** | 9.3 | 8.8 | 10.5 | 9.0 | 9.6 | 6.5 | 7.2 | 8.9 |
| | AdvDiff-Untarget | 73.1 | 70.1 | 74.9* | 73.1 | 73.9 | 69.6 | 71.2 | 64.9 | 68.0 | 71.5 | 70.6 |
| | **APA-SG(Ours)** | 75.3 | 70.8 | 99.2* | 80.4 | 80.5 | 65.0 | 63.0 | 48.1 | 42.9 | 56.3 | 64.70 |
| | **APA-GC(Ours)** | **76.6** | **77.3** | 90.2* | **82.9** | **84.0** | **70.0** | 69.7 | 53.6 | 49.5 | 63.9 | 69.72 |
| ViT-B | AdvDiff | 8.7 | 9.5 | 8.1 | 7.6 | 6.7 | 10.0 | 9.3 | **100.0*** | 8.2 | 6.9 | 8.33 |
| | AdvDiff-Untarget | 15.3 | 17.5 | 16.1 | 17.4 | 15.6 | 19.3 | 21.3 | 57.7* | 22.1 | 17.6 | 18.02 |
| | **APA-SG(Ours)** | 49.7 | 52.9 | 51.1 | 52.7 | 52.6 | 56.0 | 54.2 | 98.9* | 59.3 | 47.2 | 52.86 |
| | **APA-GC(Ours)** | **56.3** | **59.7** | **59.5** | **60.4** | **59.3** | **63.5** | **62.0** | 89.8* | **68.0** | **56.9** | **60.62** |

# F  Flexibility and Scalability

## F.1  ControlNet

To evaluate the flexibility of our framework, we utilize the Canny and Hed versions of ControlNet++[39]. As shown in Figure 10, under the constraints imposed by ControlNet, our attack optimization primarily targets texture information outside the contour regions, leaving the overall outline intact. Additionally, Table 10 highlights that adversarial examples generated using diffusion models with ControlNet maintain strong transfer attack performance.

## F.2  Extend to Other Tasks

To assess the flexibility of our method, we evaluate APA's performance across various tasks. Thanks to its high adaptability, transferring APA to new tasks requires only adjusting the corresponding $R_a$.

**Targeted Attacks.** We evaluate the feasibility of targeted attacks by specifying a target class during attack optimization. Our findings indicate that the explicit designation of the target class in the optimization process leads the generated unrestricted adversarial examples to subtly incorporate features of the target class. For instance, as shown in the right panel of Figure 11(a), when the target class is set to "hen," the generated image preserves the overall structure of the original input (ensured by our visual consistency alignment) while discreetly embedding patterns resembling hen feathers into the texture of a rock. Overall, the targeted adversarial examples generated by our method bear a resemblance to camouflaged representations of the target class.

Table 10: Attack performance on ControlNet, RN-50 as the substitute model. APA-GC-C denotes that APA utilizes a diffusion model equipped with ControlNet, with gradient propagation implemented through gradient checkpointing.

| Attack | Models | | | | | | | | | | Avg. ASR (%) |
|---|---|---|---|---|---|---|---|---|---|---|---|
| | CNNs | | | | | | Transformers | | | | |
| | MN-v2 | Inc-v3 | RN-50 | Dense-161 | RN-152 | EF-b7 | MobViT-s | ViT-B | Swin-B | PVT-v2 | |
| APA-GC-C (Hed) | 96.4 | 92.0 | 100.0* | 97.3 | **97.9** | 88.6 | 88.3 | 71.8 | 59.6 | 81.8 | 85.96 |
| APA-GC-C (Canny) | **97.6** | **92.7** | 100.0* | **97.5** | 97.6 | **90.4** | **90.4** | **75.5** | **62.5** | **84.1** | **87.58** |

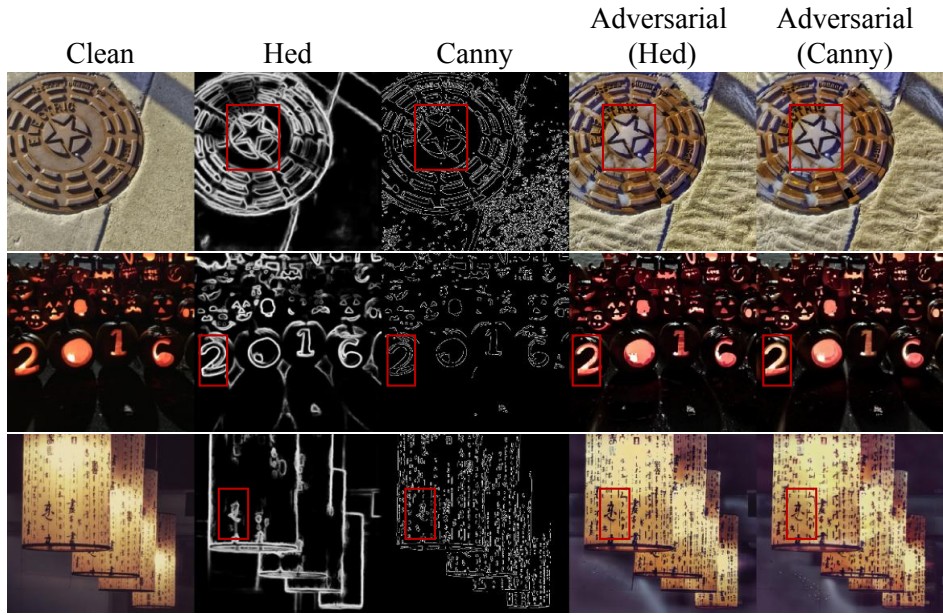

Figure 10: Visualization of adversarial examples generated by ControlNet++. Adversarial examples generated under the constraints of ControlNet can preserve the complex structures of clean images, such as text.

**Object Detection.** We further evaluate the attack effectiveness on object detection tasks, as illustrated in Figure 11(b), using $R_a$ defined as the loss function from [4]. Our results demonstrate that unrestricted adversarial examples can effectively compromise object detectors by either completely preventing object detection or inducing misclassification of detected objects.

**Visual Question Answering.** We also investigate targeted attacks on Vision-Language Models (VLMs) [50, 49]. The goal of this task is to ensure that, given a specified target text, the model generates the content of the target text when presented with adversarial examples. Figure 11(c) presents examples of unrestricted adversarial images for VLMs generated using APA. The design of $R_a$ is based on [82], where an image is first synthesized from the target text, and the cosine similarity between the features of the synthesized image and the adversarial example is computed using the BLIP-2 image encoder [38]. Our findings indicate that, consistent with the observations for targeted attacks, the generated unrestricted adversarial examples guide the diffusion model to subtly incorporate visual features corresponding to the target text into non-salient regions of the image. This enables the VLM to output the target text while disregarding the primary original objects in the image.

## F.3 Integration with different $\varrho(\cdot)$

We further investigate whether stronger data augmentation strategies can enhance the performance of our diffusion augmentation. Specifically, we adopt the data transformation methods used in SIA [72] as augmentation strategies $\varrho(\cdot)$, while keeping all other components unchanged. As shown in Table 11, the results are consistent with our findings under $L_p$ attacks: stronger data transformations further

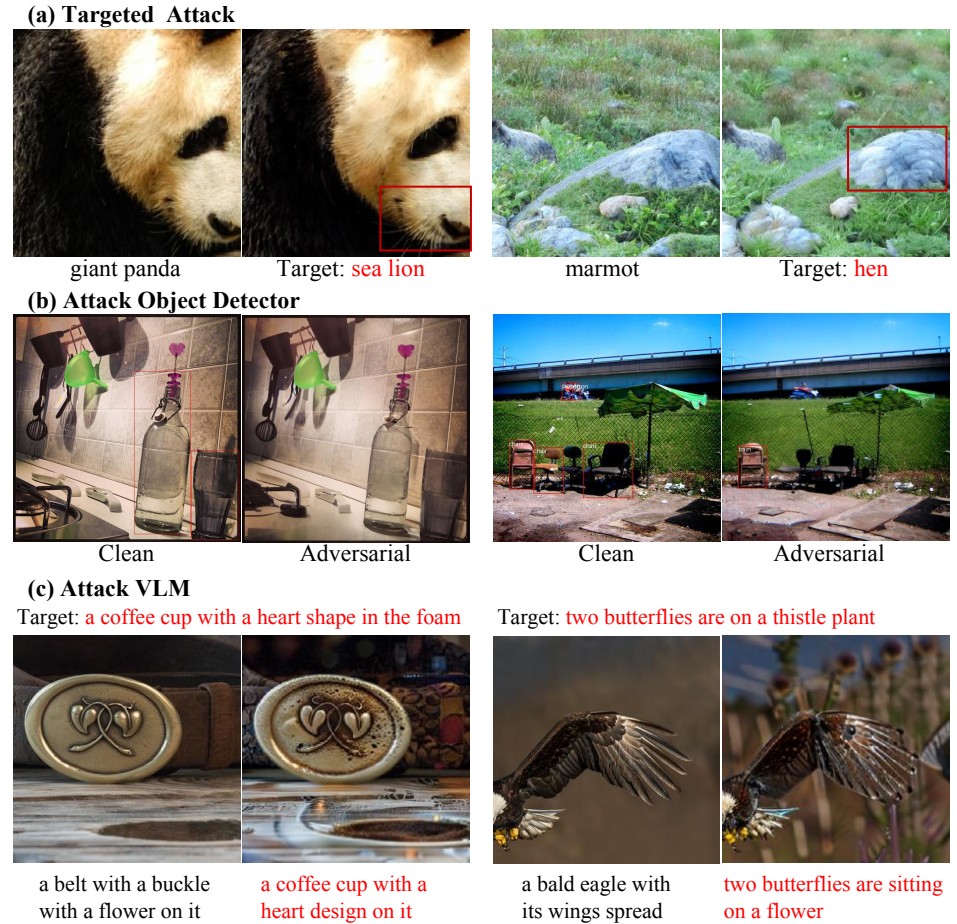

**(a) Targeted Attack**

giant panda | Target: sea lion | marmot | Target: hen

**(b) Attack Object Detector**

Clean | Adversarial | Clean | Adversarial

**(c) Attack VLM**

Target: a coffee cup with a heart shape in the foam | Target: two butterflies are on a thistle plant

a belt with a buckle with a flower on it | a coffee cup with a heart design on it | a bald eagle with its wings spread | two butterflies are sitting on a flower

Figure 11: (a) shows targeted attacks on the classification task based on RN-50 model. (b) shows untargeted attacks on the object detect task based on DETR. (c) shows targeted attacks on the visual question answering task based on VLM model BLIP-2.

Table 11: Attack performance on different $\varrho(\cdot)$. RN-50 as the substitute model.

| Attack | Models | | | | | | | | | | Avg. ASR (%) |
|---|---|---|---|---|---|---|---|---|---|---|---|
| | CNNs | | | | | | Transformers | | | | |
| | MN-v2 | Inc-v3 | RN-50 | Dense-161 | RN-152 | EF-b7 | MobViT-s | ViT-B | Swin-B | PVT-v2 | |
| APA-SG | 89.0 | 83.4 | 99.6* | 89.6 | 90.1 | 77.3 | 76.7 | 58.5 | 45.7 | 67.6 | 75.32 |
| APA-SG-SIA | **90.8** | **83.4** | **99.9*** | **91.2** | **92.6** | **80.3** | **82.9** | **62.9** | **52.7** | **72.9** | **78.86** |

improve the transferability of our framework. This demonstrates the scalability and extensibility of our framework.

# G   Visualization

We select the top six attack methods based on their performance in Table 8 and Table 1 of the main body for further visual comparison experiments, as shown in Figure 12.

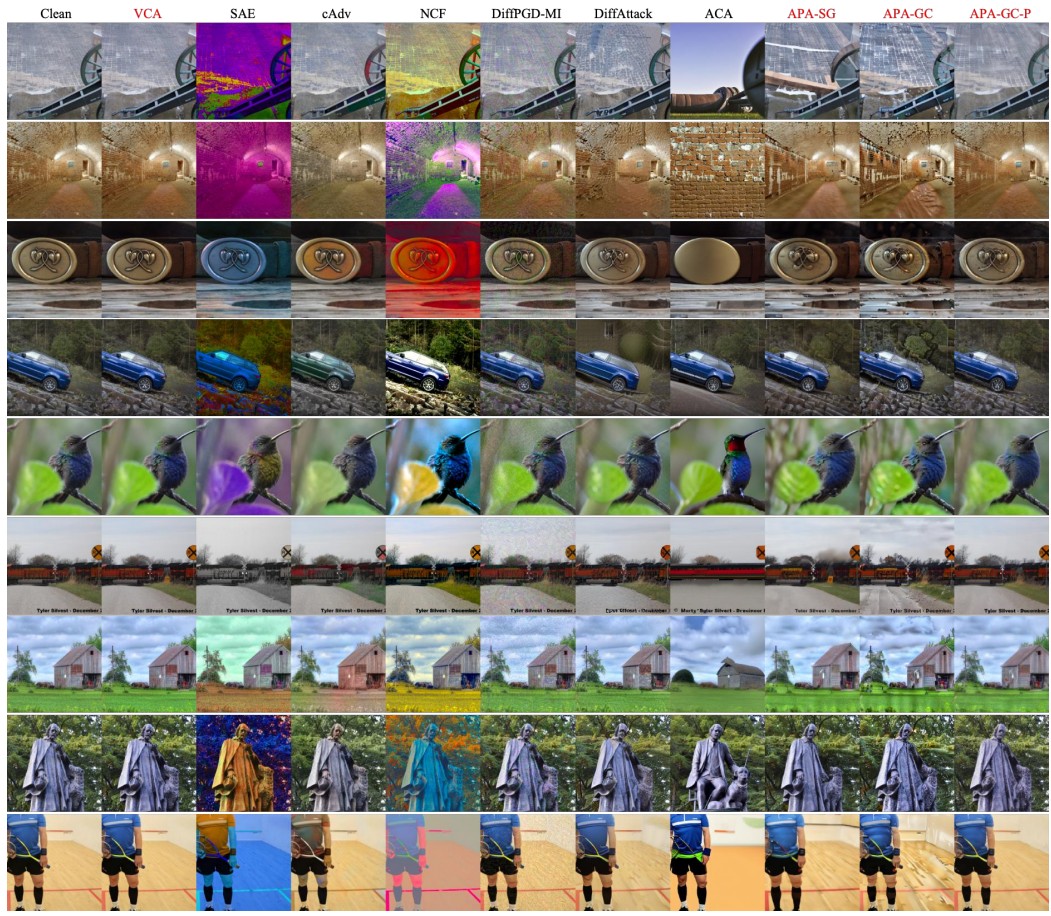

Figure 12: Qualitative comparison of image quality. Images are generated by RN-50.

