# OpenReview forum: "Enhancing Diffusion-based Unrestricted Adversarial Attacks via Adversary Preferences Alignment"
_NeurIPS.cc/2025/Conference — NeurIPS 2025 poster_

### Official Review · Reviewer_KeyC · 2025-06-26

**Clarity:** 2
**Significance:** 2
**Originality:** 2
**Rating:** 4
**Confidence:** 2

**Summary:**

The paper introduces APA (Adversary Preferences Alignment), a novel two-stage framework for generating unrestricted adversarial examples using diffusion models. The key contributions are:
1. Reformulating adversarial attacks as a preference alignment problem with conflicting objectives (visual consistency vs. attack effectiveness);
2. A decoupled approach: Stage 1 uses LoRA fine-tuning with rule-based rewards for visual consistency, Stage 2 uses dual-path attack guidance with diffusion augmentation for attack effectiveness.

**Questions:**

- How do we get equation 6 (page 4 line 168), what is the definition of
$E_{t, \epsilon}$? I think the loss function should be

$$\mathbb{E}_{t \sim [1, T], \epsilon \sim \mathcal{N}(0, \mathbf{I})} \left( - \| \epsilon - \epsilon_{\theta + \Delta \theta}(z_t, t, c)\|^2 \right).$$

It is a little ambiguous to $(E_{t,\epsilon}) - (\| \epsilon - \epsilon_{\theta + \Delta \theta}(z_t, t, c)\|^2)$.
- In equation (7), why do we use the L1 norm on $g_{tr}$? As the same norm is used in ACA, I am wandering what is the purpose of using L1 norm instead of L2 norm.
- The manuscript claims that the proposed method can achieve closer Pareto optimality, but the detail discussion is missing. Please provide more details.
- Since it is not clear whether one image is applied or a batch of images is applied to the purposed method. Various thanks for providing the code. After reading the code provided in the attachment, it seems that you fine-tune one LoRA for each Image, and attack one image using the pretrained LoRA on that Image. Does my understanding correct? I think this is horrible.
  - It is super computational expensive on getting one adversarial example.
  - It is not clear whether the fine-tuned LoRA on one image can work on other images.
  - I am not sure weather the good performance is due to something like over-fitting, and the model is highly optimized for one specific image.
- The manuscript had discussed the one-stage and two-stage alignment in page 9 line 320 to 326. It is not clear the performance of the alignment without LoRA. Please consider providing more details.

**Ethical Concerns:**

["NO or VERY MINOR ethics concerns only"]

**Final Justification:**

All my questions are clearly answered.

**Limitations:**

yes

**Quality:**

2

**Strengths And Weaknesses:**

**Strengths:**
- Novel on consider both human preference alignment and adversarial preference alignment.
- Rich experiments on various attacks and models.

**Weaknesses:**
- Heavy computation cost on per-image LoRA fine-tuning.
- Limited discussion on why the proposed method can achieve closer Pareto optimality.
- Some core operations are from references papers. For example, equation (7) is following ACA, and skip gradient also mentioned in ACA, and gradient checkpointing is widely used in network tuning. Although these are correctly cited, it would raise the concern of novelty.

---

> ### Author Rebuttal · Authors · 2025-07-29
>
> We sincerely thank the reviewer for their constructive feedback. Below, we provide point-by-point responses to each of the comments.
>
> ---
>
> **Weakness 1 & Question 4:**
> *1. Heavy computation cost on per-image LoRA fine-tuning.*
>
> *2. It is not clear whether the fine-tuned LoRA on one image can work on other images.*
>
> *3. I am not sure weather the good performance is due to something like over-fitting, and the model is highly optimized for one specific image.*
>
>
> Thank you for the reviewer’s detailed questions. **We would like to emphasize that our ultimate goal is to generate adversarial examples with strong transferability and high visual consistency. The use of LoRA serves merely as a means to achieve visual consistency alignment.** To address your concerns comprehensively, we provide clarifications from the following four perspectives:
>
> **1). Regarding the computational cost of per-image LoRA fine-tuning**
> a. Visual consistency alignment (VCA) is relatively lightweight, we update only a very small number of LoRA parameters (<1% of full parameters ), while keeping the backbone model frozen.
>
> b. On an A100 GPU, fine-tuning for a single image takes approximately **38 seconds**. In comparison, the null-text optimization used in the previous state-of-the-art **ACA** method takes around **55 seconds**.
>
> c. VCA only requires around **6GB** of GPU memory, making it a lightweight module with minimal resource demands.
>
> Overall, we believe that the computational cost is acceptable given the significant performance VCA gains (please see R#a1PE Question 1).
>
> **2). Regarding whether the fine-tuned LoRA can transfer to other images**
> Our method is fundamentally an **image-specific adversarial attack framework**, which optimizes a fine-tuning module (LoRA) for a given input image to preserve its high-level semantics.  As such, we do **not** aim for generalization across entirely different images.
>
> Under this setting, LoRA is inherently **tailored** to each image and is **not intended to be transferable** to arbitrary inputs — which aligns with our design objective.
>
> Nevertheless, we believe that good transferability should exist among **semantically similar images**. To validate this, we conducted the following experiment:
>
> - We randomly selected 5 video clips from **UCF101** video dataset.
> - From each clip, we extracted **16 consecutive frames**.
> - We trained a LoRA on the **first frame** of each clip using our **VCA**.
> - Then we applied **the same LoRA** to subsequent frames to evaluate its reconstruction ability on semantically similar images.
>
> The results are shown below:
>
> | Type| LPIPS↓ | SSIM↑ | CLIP Score↑ |
> |-------------------------------------|--------|-------|--------------|
> | First frame w/ first frame VCA  | 0.05   | 0.91  | 0.96 |
> | Whole video w/ first frame VCA | 0.06   | 0.90  | 0.94|
> | Whole video w/o first frame VCA | 0.21   | 0.79  | 0.81|
>
> We observe that the LoRA fine-tuned on the first frame generalizes well to the subsequent frames, suggesting that our VCA module exhibits good generalization across images with similar semantics. This observation indicates a potential future optimization: grouping batch data by semantic similarity and performing VCA once per group, rather than per image.
>
>
> **3) Regarding concerns about “overfitting”**
>
> We believe this concern stems from a misunderstanding: that overfitting the LoRA parameters to a single image would compromise the generalization of the attack.
>
> In fact, our method fine-tunes the LoRA module in an image-specific manner to preserve semantic consistency with the input. The generalization, however, arises from the adversarial image itself, not from the LoRA parameters.
>
> Specifically, we first align visual semantics via LoRA fine-tuning, then apply an alignment loss targeting the victim model to improve transferability. This two-stage strategy ensures both semantic preservation and strong cross-model attack success (see Table 1). Thus, while the optimization is per-image, the resulting adversarial image generalizes well across models.
>
>
>
> **4) On the advantages and justification of image-specific optimization**
>
> **Image-specific optimization** is a widely adopted strategy in both adversarial attacks and personalized generation. For instance, many diffusion-based attack methods and personalized generation[1] fine-tune models per image or subject to ensure high fidelity.
>
> Compared to dataset-level optimization, image-specific tuning offers key advantages:
>
> - **Flexibility:** A single alignment can generalize across multiple downstream tasks (see Appendix G.2).
> - **Higher performance ceiling:** Tailoring to each image enables better use of its semantic features.
> - **Faster convergence:** More stable and efficient without large-scale data.
> - **Low adaptation cost:** Less dependent on task/model configurations than dataset-level methods.
>
> [1]DreamBooth: Fine Tuning Text-to-Image Diffusion Models for Subject-Driven Generation, CVPR, 2023.
>
> ---
>
> **Weakness 2 & Question 3:**
> *Limited discussion on why the proposed method can achieve closer Pareto optimality.*
>
>
> Thank you for your insightful comment. In multi-objective optimization, the Pareto optimal set consists of solutions where no objective can be improved without sacrificing another. Since the true Pareto frontier is intractable in our task, we construct the analysis shown in **Figure 5(b)** to approximate it. In this figure, the x-axis measures **visual consistency**, and the y-axis represents **attack effectiveness**. Ideally, Pareto-optimal solutions lie toward the **top-right corner**, where both objectives are maximized.
>
> Our proposed algorithm, **APA-SG** (red points), first approaches the optimal solution space for visual consistency and then searches locally for solutions with higher attack effectiveness. Compared to other joint optimization methods (blue points), our approach yields results that are **closer to the Pareto-optimal frontier** (i.e., the upper-right region of the plot).
>
> We will provide a more detailed discussion of this insight in the **final version**, specifically in the analysis of **Section 5.5**.
>
> ---
>
> **Weakness 3:**
> *Some core operations are from references papers. For example, equation (7) is following ACA, and skip gradient also mentioned in ACA, and gradient checkpointing is widely used in network tuning. Although these are correctly cited, it would raise the concern of novelty.*
>
>
> Thank you for your comments. We would like to clarify that we never claimed these two gradient computation methods as our own contributions. These are standard components in attack optimization, much like how backpropagation is standard in model training. **Our intention in describing these methods is to highlight the flexibility of our framework, which can readily incorporate different gradient computation strategies as needed.**
>
> While previous works have adopted these optimization techniques, none has systematically examined how different gradient computation strategies affect the attack performance — a gap our work addresses.
>
> Most importantly, our main innovations are as follows:
>
> (1) For the first time, we interpret the unrestricted adversarial example generation problem from the perspective of preference alignment.
>
> (2) We decouple  the conflicting preferences and propose a two-stage alignment framework that is highly flexible and adaptable to different tasks, diffusion models, optimization parameters.
>
> (3) To address the unique needs of attack preference (i.e., strong attack generalization), we introduce dual-path attack optimization and diffusion augmentation, which improve attack preference alignment and prevent reward hacking.
>
> In summary, the core novelty of our work lies in the **new perspective**, **the general and flexible optimization framework**, and **the newly designed components for attack preference alignment**—not in the specific gradient calculation methods themselves.
>
> ---
>
> **Question 1:**
> *Correction of Eq. 6.*
>
> Thank you for your insightful comment. You are absolutely right — since a similar form was already introduced in Equation 2, we applied a simplified notation in Equation 6.  As per your suggestion, we will revise the equation accordingly. We sincerely appreciate your feedback.
>
> ---
>
> **Question 2:**
> *In Equation (7), why do we use the L1 norm*
>
> Thank you for your careful observation. Both ACA and our method follow the classical work in adversarial attacks — MI [1], which first introduced momentum into adversarial example generation. As stated in the original paper (last paragraph of Section 3.1), any distance metric can be used for normalization, with L1 norm being the default. Therefore, using the L2 norm is also valid.
>
> [1] Dong et al., *Boosting Adversarial Attacks with Momentum*, CVPR 2018.
>
> ---
>
> **Question 5:**
> *The manuscript had discussed the one-stage and two-stage alignment in page 9 line 320 to 326. It is not clear the performance of the alignment without LoRA. Please consider providing more details.*
>
> Thank you for your valuable comment.  Following your suggestion, we have now included the corresponding quantitative results, shown in the table below, ResNet50 as the source model:
>
> | Method| LPIPS↓ | SSIM↑ | CLIP Score↑ |  Avg. ASR
> |---------------------|--------|--------|----------------| ----------------|
> | APA-SG w/o VCA | 0.55| 0.46| 0.62|  95.43 |
> | APA-SG| 0.25| 0.67| 0.86|75.32|
>
> These results show that aligning only with attack effectiveness significantly boosts attack performance, but severely degrades image quality (CLIP Score = 0.62), making the outputs practically unusable. We will add these results to Figure 5(b) and discuss them on page 9, lines 320–326.
>
> Once again, thank you for your time. If these clarifications resolve your concerns, we would be grateful if you could consider raising your score.

---

> > ### Comment · Reviewer_KeyC · 2025-08-02
> >
> > **per-image LoRA fine-tuning**:
> > - If your method is work on per-image, this key information **must** be clearly transfer to the reader in the paper, instead of making the reader guessing from your code. As it is very different from other attack methods.
> > - If your method is based on fine tuning a model for each image, then in experiment, the same set of methods should be compared, i.e., also fine tune a model for a specific image to attack. Please specify which comparing method also has this kind of setting.
> > - It seems that ACA has the closest setting (it learned in inputs, but your method is in weights). I am curious that does the null text embedding $\phi$  learn from one images in ACA works on that specific image or can work on other images? (i.e., reusable of the $\phi$)
> > - The experiment in "Weakness 1 & Question 4, 2)" using video data. I think it is able show your method can transfer to other images with tiny changes, since the distances between nearby frames in video is usually tiny. However, it cannot shows its general capability of transfer to arbitrary images. Since your method is per image specific, I think it is acceptable. Please make this clear in your paper.
> > - About the concern on "overfitting". If the paper is clearly claim that the method train LoRA with respect to one image, I won't have this concern.
> > - The paper "DreamBooth" train of a subset (3~5) of similar images, and it is not for attacking. But I don't think this is a problem.
> > - Why in answer to Reviewer sWRo, 3090 is used, and in this answer A100 is used? (Just curious. May be just picking a free environment. Please ignore this question if you don't want to reply this question.)
> >
> > **All the rest**: It is clear to me.

---

> > > ### Author Response · Authors · 2025-08-02
> > >
> > > Thank you for your detailed comments and constructive feedback. Before addressing your points individually, we would like to clarify two key aspects:
> > >
> > > 1. **Our primary goal is to generate high-quality, unrestricted adversarial examples, rather than to train a generalizable model.** Any attempt to generalize LoRA across multiple images is solely for accelerating batch processing and comes at much higher training difficulty compared to per-image optimization.
> > >
> > > 2. **Per-image optimization is standard practice in adversarial example literature.** All baseline methods in Table 1 are based on this paradigm; this is not unique to our work.
> > >
> > > **We will explicitly highlight the image-specific alignment setting of our method in both the “Method” section (under Visual Consistency Alignment) and the Experimental Setup to ensure full clarity.** We sincerely appreciate your suggestions for making this more prominent.
> > >
> > >
> > >
> > > ---
> > >
> > > **A1.** Thank you for pointing this out. Our method indeed employs per-image optimization, which is a common setting in adversarial attack literature and is also adopted by all baselines in Table 1. We will explicitly state this in both the Method and Experimental Setup sections to prevent any ambiguity. We also acknowledge that LoRA-based optimization rarely adopts this setting and will emphasize this difference in our revision.
> > >
> > >
> > >
> > > **A2.** The disentangled visual consistency alignment is one of our key innovations. While ACA’s null-text optimization plays a related role, as shown in Table 3, its performance is inferior to ours. Moreover, our further analysis of ACA reveals that its visual consistency mainly relies on the MSE loss used during attack optimization. Additionally, when conducting ablations on the *dual-attack optimization* and *diffusion augmentation* in our *attack effectiveness alignment*, we ensure the same *visual consistency alignment* is applied for fair comparison and to highlight the contribution of each module in Figure 5(a) and Table 4.
> > >
> > > **A3.** No, the null-text embedding learned in ACA is also specific to each image and is not reusable across different images.
> > >
> > >
> > > **A4&A5.** We will explicitly highlight the image-specific optimization setting of our method in both the “Method” section (under Visual Consistency Alignment) and the Experimental Setup to ensure full clarity. We sincerely appreciate your suggestions for making this more prominent.
> > >
> > >
> > >
> > > **A7.** This was done solely to align with the experimental setting of ACA. Device differences have no impact on our performance.
> > >
> > >
> > > Once again, thank you for your time. If these clarifications resolve your concerns, we would be grateful if you could consider raising your score.

---

> > > > ### Comment · Reviewer_KeyC · 2025-08-02
> > > >
> > > > Thank you for clarifying.
> > > > Now all my questions are clearly answered.
> > > > Please keep your promise and make it clear to readers.

---

> > > > > ### Author Response · Authors · 2025-08-02
> > > > >
> > > > > Thank you for your valuable contribution to our work. We will revise the final version accordingly based on your suggestions. Once again, we sincerely appreciate your feedback.

---

### Official Review · Reviewer_ELVx · 2025-06-28

**Clarity:** 2
**Significance:** 3
**Originality:** 2
**Rating:** 4
**Confidence:** 5

**Summary:**

This paper proposes APA, a novel framework reframing unrestricted adversarial example generation in diffusion models as alignment with adversary preferences. Addressing the conflicting objectives of visual consistency and attack effectiveness – which often cause unstable optimization and reward hacking – APA employs a two-stage approach: first fine-tuning LoRA for visual quality using rule-based rewards, then optimizing image latents or prompt embeddings using substitute classifier feedback with trajectory-level and step-wise rewards. Enhanced by diffusion augmentation for transferability, APA achieves significantly better black-box attack success while maintaining high visual consistency.

**Questions:**

1. While the approach presents an interesting exploration of unrestricted attacks, the connection to adversary preference alignment appears tenuous. Consequently, the paper's organization makes it difficult to grasp the necessity of the numerous proposed techniques. The authors are adviced to **significantly improve the presentation clarity**. For reference, my understanding is that the attack comprises two core modules:

  - Optimizing $z_T$ for attack effectiveness. Since diffusion model is instable near $z_T$, LoRA-based training is used to enhance visual consistency.

  - Solving an inverse problem (maximizing adv. loss) via denoising path guidance, similar to DPS[1].

2. Although APA-GC demonstrates the best performance, the authors provide no analysis of its computational efficiency. It is crucial to compare the performance of different attack algorithms under equivalent computational budgets. For example, the authors can include results using standardized metrics, such as ASR plotted against computation time (e.g., wall-clock time or FLOPs), with time/compute on the x-axis and ASR on the y-axis.

[1] Diffusion Posterior Sampling for General Noisy Inverse Problems. (https://arxiv.org/abs/2209.14687)

**Ethical Concerns:**

["NO or VERY MINOR ethics concerns only"]

**Final Justification:**

The clarity of the presentation and the time overhead of the methods are not ideal, but this is not a fatal flaw.

**Limitations:**

Yes.

**Quality:**

3

**Strengths And Weaknesses:**

Strengths:

1. This paper presents APA, an effective framework addressing the fundamental trade-off between visual consistency and attack effectiveness in adversarial example generation.
2. The experiments are thorough and well-designed, providing convincing evidence for the algorithm’s effectiveness and superiority.

Weaknesses：

1. The paper’s organization is challenging to follow, with an abundance of complex design choices that obscure the core methodology and hinder readability.
2. The framework’s high time overhead significantly limits its real-world applicability.

---

> ### Author Rebuttal · Authors · 2025-07-30
>
> We sincerely thank the reviewer for their constructive feedback and for recognizing the effectiveness of our method and the thoroughness of our experiments. Below, we provide point-by-point responses to each comment.
>
> ---
>
> **Weakness 1 & Question 1.2:**
> *The paper’s organization is challenging to follow, with an abundance of complex design choices that obscure the core methodology and hinder readability. Consequently, the paper's organization makes it difficult to grasp the necessity of the numerous proposed techniques. The authors are adviced to significantly improve the presentation clarity.*
>
> We thank the reviewer for the feedback. We will revise the manuscript to emphasize how each serves a distinct role in the alignment process, and de-emphasize optional parts to reduce perceived complexity, primarily in the following aspects:
>
> 1. **Clarifying the core methodology**
> We will restructure Section 4 to clearly present our two-stage alignment framework at the outset and add a concise overview summarizing the core pipeline.
> - **Stage 1** aligns visual consistency by encoding the input image into LoRA parameters.
> - **Stage 2** enhances attack transferability through dual-path optimization and diffusion augmentation.
>
> 2. **Reducing perceived complexity**
> While our framework includes several components, we will clearly distinguish between core modules (e.g., alignment stages, dual-path attack optimization) and optional extensions (e.g., different optimization parameters, gradient strategies). To improve readability, we will include a brief summary of these in the paragraph starting at line 230.
>
> 3. **Highlighting simplicity and extensibility**
> Our framework is easy to implement with minimal modifications to Hugging Face’s Diffusers library. It is decoupled from specific models and tasks, making it generalizable to a wide range of downstream applications.
>
> We believe these revisions will significantly improve the paper’s clarity and accessibility.
>
> ---
>
> **Weakness 2:**
> *The framework’s high time overhead significantly limits its real-world applicability.*
>
>
> We thank the reviewer for the concern regarding time cost, which we have acknowledged as a limitation in our paper. However, we would like to clarify two key points. First, our method offers a relative efficiency advantage over existing approaches. Second, in current applications of unrestricted adversarial examples, the primary focus lies in attack performance and image quality, with efficiency being a secondary yet desirable attribute. We further elaborate on this issue from four perspectives:
>
> **1. Competitive Efficiency Compared to SOTA**
>
> As reported in Appendix B, our method achieves approximately **16% speedup** over the SOTA method ACA on a single A100 GPU, while significantly outperforming ACA in terms of **attack success rate and visual quality** (see Tables 1,2,3). This demonstrates a favorable trade-off between efficiency and performance.
>
>
> **2. Real-World Applicability Is Primarily Offline**
>
> Unrestricted adversarial examples serve two key real-world purposes: **(1) as a tool for adversarial robustness evaluation, particularly under transfer-based attacks;** and **(2) as hard samples for data augmentation（recent studies[1,2] have shown that hard samples generated by diffusion models can substantially improve downstream model performance when used for training)**. Importantly, both applications are conducted offline and thus impose no strict real-time constraints on the generation process.
>
> In summary, the practical value of unrestricted adversarial examples lies in their ability to stress-test and enhance models. Their generation time is largely irrelevant in these contexts, as performance—not speed—is the primary concern.
>
>
> [1]ASAM: Boosting Segment Anything Model with Adversarial Tuning, CVPR,2024
>
> [2]Synthesizing Near-Boundary OOD Samples for Out-of-Distribution Detection, ICCV,2025
>
> **3. Acceleration Strategies**
>
> The main overhead stems from attack effectiveness alignment, particularly the attack iterations and denoising steps during diffusion sampling. Thanks to the strong performance of our method, we can further reduce the number of attack iterations to improve efficiency (see Question 2 for details). Moreover, more efficient sampling strategies may offer further acceleration opportunities.
>
> We have implemented a parallel version of our pipeline to support **batch-wise optimization** for attack alignment, substantially improving the efficiency of generating batched adversarial examples. The updated code will be released upon paper acceptance.
>
>
>
>
> ---
>
> **Question 1.1:**
> *While the approach presents an interesting exploration of unrestricted attacks, the connection to adversary preference alignment appears tenuous.*
>
> Thank you for your comments. We would like to clarify that the key distinction between our APA and prior unrestricted attacks lies in our **alignment-driven modeling**. Specifically, our framework is motivated by the need to:
>
> - define malicious preferences (*visual consistency* and *attack effectiveness*),
> - quantify these preferences via *rule-based* or *learned* reward functions,
> - select appropriate alignment strategies (*DPO*, *reinforcement learning*, or *direct backpropagation*),
> - define stable alignment framework (*joint* or *decouple*)
> - design task-specific modules such as *dual-path optimization* and *diffusion augmentation*.
>
> These components are unified in our proposed two-stage alignment framework, where the central novelty lies in **decoupling the alignment of inherently conflicting preferences**—i.e., maximizing attack performance within the solution space constrained by visual consistency, thereby approaching closer to the Pareto frontier. Furthermore, *dual-path optimization* and *diffusion augmentation* are specifically introduced to mitigate overfitting issues commonly observed in direct backpropagation. We validate the effectiveness, flexibility, and robustness of our method through extensive experiments.
>
> ---
>
> **Question 1.3:**
> *Solving an inverse problem (maximizing adv. loss) via denoising path guidance, similar to DPS.*
>
> Thank you for your suggestion. We carefully compared DPS with our method and found they address fundamentally different goals. DPS solves an inverse problem focused on reconstruction quality, whereas our attack alignment targets the generation of transferable adversarial examples, where reconstruction quality is already ensured by visual consistency alignment.
>
> The only loose connection lies in using gradient-based guidance: DPS leverages reconstruction loss gradients, while our dual-path guidance is tailored for adversarial alignment. **Notably, such guidance originates from classifier-guided diffusion, which we already compare against—along with its stronger variant, Upainting—both being conceptually closer to our method than DPS.**
>
> As shown in Figure 5(b), our dual-path guidance, incorporating a cleaner latent reference ($z_{in}$ in Eq. 10) and momentum accumulation, yields significantly better attack performance. We will include a discussion on DPS in the final version.
>
> ---
>
> **Question 2:**
> *Although APA-GC demonstrates the best performance, the authors provide no analysis of its computational efficiency. It is crucial to compare the performance of different attack algorithms under equivalent computational budgets. The authors can include results using standardized metrics, such as ASR plotted against computation time (e.g., wall-clock time or FLOPs), with time/compute on the x-axis and ASR on the y-axis.*
>
>
> Thank you for your suggestion. First, we have already analyzed the time efficiency of our method compared to the state-of-the-art ACA in Appendix B. Under the same number of attack iterations and denoising steps, our method achieves **higher efficiency** and **significantly better performance** than ACA.
>
> We fully agree with your point. As discussed earlier, the primary time cost of alignment-based attacks stems from the attack optimization and the diffusion denoising process. **Since the number of attack iterations and denoising steps directly determines the number of forward/backward passes through the UNet, they serve as reliable proxies for overall compute cost.** We therefore analyze both components separately to explore whether efficiency can be further improved without compromising performance.
>
> **1. Attack Iteration**
>
> Although we are unable to include images and figures, we report the transfer success rate of our method across different numbers of attack iterations (see the table below). For reference, ACA achieves **59.49 Avg.ASR**.
> | Iterations| 1| 2| 3| 4| 5| 6| 7| 8| 9| 10|
> |----------|-------|-------|-------|-------|-------|-------|-------|-------|-------|-------|
> | APA-SG| 27.96| 37.73| 44.62| 51.26| 57.42| 62.25| 66.03| 69.53| 72.48| 75.32|
> | APA-GC|  25.56|  40.22|    51.82|   61.09|    68.86|   74.88|   79.83|   83.76|   87.01|    88.02
>
> Our **APA-SG** surpasses this benchmark within just **6 iterations**, and **APA-GC** within only **4**. This demonstrates that, thanks to the superiority of our approach, we can significantly reduce the number of iterations while still achieving state-of-the-art attack performance. **In the final version, we will convert the above table into a curve plot.**
>
> **2. Denoising Process**
>
> As shown in Figure 6(a) and 6(c) of the main paper, the number of denoising steps can also be further reduced. For example, using only **5 denoising steps** (Figure 6a), our method achieves a **1.7× speedup** while still reaching a success rate of **87.14**, which remains substantially higher than ACA.
>
> Once again, thank you for your time. We sincerely hope our response can address your concerns.

---

> > ### Comment · Reviewer_ELVx · 2025-08-02
> >
> > I don't think you need to compare yourselves to DPS, as this clearly doesn't align with your goals. My suggestion is that explaining your work from the perspective of inverse problem sampling (e.g., let $p(y|x) \propto e^{−R(x)}$, where $R$ is the reward function) rather than reinforcement learning might be more appropriate. If you believe your work is closer to a reinforcement learning problem, I would like you to describe the environment, actions, and other elements. And as you mentioned (Answer to Q1.1 point 3), why not use DPO or reinforcement learning?

---

> > > ### Author Response · Authors · 2025-08-02
> > >
> > > Thank you very much for your fast and helpful feedback and suggestions.
> > >
> > > First, our previous response to Question 1.3 focused more on technical details. We agree that viewing the unrestricted adversarial example generation problem as an inverse problem is a valuable perspective, i.e., Given a clean sample $x$, using $\nabla_{x_t} (R_a+R_s)$ to guide the generation towards both visual consistency and attack effectiveness. With our decoupling design, visual consistency is handled in the first stage through LoRA optimization, while attack effectiveness becomes a single-objective inverse problem in the second stage. We will include this inverse problem viewpoint in the appendix.
> > >
> > > As mentioned in our response to Question 1.1, the motivation for our work stems from an intriguing question: Can diffusion models be aligned with "malicious" preferences? This is essentially a preference alignment problem, where the ultimate goal is to maximize defined reward. Reinforcement learning (RL) is one way to achieve reward maximization. Previous work [1] modeled the diffusion process as a Markov Decision Process (MDP) (with definitions of state, action, etc., as in  paper), and used Monte Carlo or PPO for policy gradient estimation. However, such RL-based approaches suffer from high variance, low efficiency, and are challenging to implement. Offline DPO is another alternative [2], but obtaining high-quality ranking pairs for adversarial attacks is costly, and DPO itself is prone to overfitting to the distribution of the training data.
> > >
> > > Instead, our method adopts a more efficient approach—direct backpropagation [3,4]. When the reward function is differentiable, policy gradients can be directly computed based on the complete gradient chain, enabling end-to-end and efficient training. This eliminates the need to explicitly define an MDP tuple (e.g. state, action ) to calculate policy gradients. This is also why we used two differentiable reward functions. However, direct optimization can sometimes lead to over-optimization (i.e., generating high reward samples  but lack attack transferability), so we propose a diffusion enhancement mechanism to alleviate this issue.
> > >
> > > Overall, from the perspective of alignment, our approach starts by considering how to align with malicious preferences. We decouple the two conflicting preferences and define differentiable rewards for each. Then, we use a direct backpropagation as efficient optimization method to achieve optimal performance for both rewards. Based on the above analysis, our approach can also be fully interpreted from the perspective of the inverse problem, as you suggested.
> > >
> > > These details are discussed further in the related work section, and we will clarify and emphasize these points in the final version as well.
> > >
> > > Once again, thank you for your time. If these clarifications resolve your concerns, we would be grateful if you could consider raising your score.
> > >
> > > [1] Training diffusion models with reinforcement learning. In ICLR, 2024.
> > >
> > > [2] Diffusion model alignment using direct preference optimization. In CVPR, 2024.
> > >
> > > [3] Directly fine-tuning diffusion models on differentiable rewards. In ICLR, 2024.
> > >
> > > [4] Textcraftor: Your text encoder can be image quality controller. In CVPR, 2024.

---

> > > > ### Comment · Reviewer_ELVx · 2025-08-02
> > > >
> > > > In LLM, the most commonly used method for preference alignment is RLHF, which is what readers are most likely to think of. Of course, by definition, preference alignment can involve neither humans nor RL, as your paper demonstrates. However, the use of this concept does indeed confuse some readers (including myself at least).
> > > >
> > > > Finally, I have no concerns about the other parts of the paper, and I tend to keep my score :) it's a good work anyway.

---

> > > > > ### Author Response · Authors · 2025-08-02
> > > > >
> > > > > Thank you very much for your positive recognition of our work. We will carefully incorporate your suggestions into our final version.

---

### Official Review · Reviewer_sWRo · 2025-07-02

**Clarity:** 2
**Significance:** 3
**Originality:** 3
**Rating:** 4
**Confidence:** 4

**Summary:**

This paper introduces the Adversary Preferences Alignment (APA), an innovative two-stage framework designed to enhance diffusion-based unrestricted adversarial attacks. Unlike typical benign preference alignments that optimize for human-preferred attributes, APA addresses adversarial preferences by balancing two conflicting goals: visual consistency and attack effectiveness. The authors propose decoupling these conflicting objectives, initially optimizing visual consistency through LoRA fine-tuning with a rule-based visual similarity reward, followed by optimizing attack effectiveness using dual-path attack guidance and diffusion augmentation.

**Questions:**

1. In the paper, you conducted evaluation on existing defenses which originally focusing on Lp attacks, but are you also considering defenses method on unrestricted adversarial attacks?

2. In Eq.7, it introduces momentum-based gradients, but $\Pi z^0_T + \epsilon_a$, what this refer to? Can you provide more details?

3. Line 164 ~166, you mentioned regarding the shift of similarity metric from the pixel/image space (Eq. 5) to the latent space (Eq. 6) for computational efficiency. However, the paper does not provide sufficient empirical comparison to support the claim that latent similarity adequately preserves visual consistency, which is critical to the proposed method. Can you possibly explain more on this or provide more context?

4.  The “skip gradient” and “gradient checkpointing” strategies are described briefly; their impact on optimization stability and convergence, I think is not well justified or analyzed. Are there convergence guarantees ?

**Ethical Concerns:**

["NO or VERY MINOR ethics concerns only"]

**Final Justification:**

- **Focus on My Original Concerns:**
  My main concerns were:
  1) Practical deployment complexity due to the multi-component structure.
  2) The need for stronger empirical justification for latent-space similarity as a surrogate for visual consistency.
  3) Improving clarity of notation and optimization flow.

  The authors’ rebuttal addressed these well: they committed to restructuring the methodology section, adding a concise pipeline overview, and distinguishing core vs. optional components to reduce perceived complexity. They also reiterated the rationale for latent-space similarity and linked it to visual consistency, while promising clearer integration in the final version. These commitments address the core of my concerns, although the improvements will only be realized in the camera-ready.

- **Consideration of Other Reviewers’ Concerns:**
  I have reviewed the other reviewers’ comments and the authors’ responses. Key issues raised — such as organization, necessity of components, computational efficiency, and the framing of the method — were all answered with detailed clarifications. The authors not only justified their design choices (e.g., direct backpropagation vs. RL/DPO) but also provided quantitative evidence for efficiency trade-offs and acknowledged the inverse-problem perspective.

- **Assessment of Authors’ Responsiveness:**
  The rebuttal was comprehensive, technically precise, and showed a willingness to integrate feedback from all reviewers. The level of detail and empirical support provided in their clarifications gives me confidence that the final version will substantially improve in clarity and presentation.

- **Remaining Issues:**
  The dense presentation of the original submission remains a limitation until the promised revisions are implemented. I also believe more explicit empirical connection between latent-space similarity and visual consistency would further strengthen the paper.

- **Recommendation Rationale:**
  Given the strong technical contributions, thorough experimental results, and the authors’ demonstrated ability to address feedback from multiple reviewers, I recommend acceptance while keeping my original score to reflect that the current submission still relies on future revisions for full clarity.

**Limitations:**

The authors have provided the limitations in Appendix C.

**Paper Formatting Concerns:**

There are no Paper Formatting Concerns.

**Quality:**

2

**Strengths And Weaknesses:**

## Strengths:

- Concept of aligning diffusion models to adversary preferences rather than benign human preferences
- I appreciate that the author conduct comprehensive evaluation across multiple CNNs and Vision Transformers
- APA significantly outperforms prior methods in maintaining visual consistency and naturalness, supported by rigorous quantitative  (LPIPS, SSIM, CLIP Score) and qualitative assessments.
- The authors provided the discussion regarding the ablation of visual consistency alignment  and gradient backpropagation.

## Weakness:
- Although the 2-stage method is novel, multi-component optimization process may introduce complexity in real-world implementation. Possibly further simplification or integration techniques could be beneficial.
- The paper frequently uses $z_T$, $z_0$, $\bar{z}_0$, $\hat{z}_0$, $z^0_t$, $z^{in}_t$, etc., sometimes with and sometimes without bars, hats, or superscripts. It's easy for the readers to lose track of what each variable specifically represents—particularly the difference between “trajectory-level” and “step-level” variables. Maybe you could provide a summary table of symbols.
- And next on equations. The reader is left piecing together the order and purpose of updates without a clear, step-by-step guide. For example:
   - In Eq. 6, the switch from image similarity to latent similarity is abrupt; justification for why latent-space similarity is a good surrogate for visual consistency is not well articulated.
- Regarding the explanation of methods and strategies, I believe you need to provide more clarification. For example, details on LoRa Fine-Tuning, to be more precise, how LoRa modules are inserted.

---

> ### Author Rebuttal · Authors · 2025-07-29
>
> We sincerely thank the reviewer for their constructive feedback and for recognizing the novelty of our method and the thoroughness of our experiments. Below, we provide point-by-point responses to each comment.
>
> ---
>
> **Weakness 1:**
> *Although the 2-stage method is novel, multi-component optimization process may introduce complexity in real-world implementation. Possibly further simplification or integration techniques could be beneficial.*
>
> Thank you for your valuable comment. We understand the concern regarding potential complexity in real-world implementation due to the multi-component design. We would like to clarify that our method is straightforward to implement and highly modular, as outlined below:
>
> **Simple and Modular Implementation:**
> Unlike prior approaches that tightly couple objectives into a single optimization loop, our method is implemented as a clean two-stage pipeline. The visual consistency alignment is based on **lightweight** LoRA fine-tuning, and after merging the LoRA parameters into the UNet, **no additional parameters are introduced for the subsequent attack alignment.** The attack alignment is seamlessly integrated into the denoising process via **plug-and-play modules** (i.e., dual-path attack optimization + diffusion augmentation). The entire pipeline is fully **compatible with the widely-used Diffusers library, requiring only a few lines of code to integrate with existing workflows.**
>
> **Low Integration Cost:**
> Despite its modularity, the default configuration works well across various settings, **requiring minimal hyperparameter tuning.** For most use cases, users only need to specify the gradient computation method (e.g., skip-step or gradient checkpointing), with the rest handled by default wrappers. **The full method runs efficiently on a single RTX 3090 GPU without special hardware or distributed training.**
>
> We will revise the paper accordingly to make the simplicity of our implementation more transparent. Specifically, **we will add a structural overview and an integration guide in the Appendix**.
>
> ---
>
> **Weakness 2:**
> *provide a summary table of symbols*
>
> Thank you for your comment. We have provided a detailed symbol table to aid your understanding.
>
> ### Symbol Table
>
> | Symbol       | Description |
> |--------------|-----------------------------------------------------------------------------|
> | $z_T$        | The latent updated at each iteration, initialized as $z_T^0$               |
> | $z_T^0$      | Latent representation obtained by applying DDIM inversion to the original image latent for T steps |
> | $z_0$        | VAE-encoded latent of the original image                                    |
> | $\bar{z}_0$  | Reconstructed $z_0$ obtained by DDIM denoising for T steps                  |
> | $z_0^t$      | Reconstructed latent based on $z_t$, as defined in Equation (9); $z_0^0 = \bar{z}_0$ |
> | $z_{in}^t$ | Interpolation between $z_0$ and $z_0^t$                                 |
> | $g_{tr}$ | Gradient at the trajectory level                                         |
> | $m_{tr}$ | Momentum at the trajectory level                                         |
> | $g_{st}$ | Gradient at the step level                                               |
> | $m_{st}$ | Momentum at the step level                                               |
>
> A more detailed description is provided in Appendix D, which can be best understood together with the accompanying pseudocode.
>
> ---
>
> **Weakness 3 & Question 3:**
> *In Eq. 6, the switch from image similarity to latent similarity is abrupt; justification for why latent-space similarity is a good surrogate for visual consistency is not well articulated.*
>
> Regarding the shift from pixel-space similarity to latent-space similarity, we appreciate your observation. **Our design follows the motivation outlined in the LDM paper[1]**, particularly the concept of perceptual compression—that is, most pixel-level information is redundant, and perceptually equivalent representations can be obtained in the latent space through a trained autoencoder (see Sec. 3.2 and Fig. 2 in the original paper). Therefore, we perform consistency optimization in the latent space, which preserves high-level semantics while improving training efficiency and stability. Although this substitution is not strictly equivalent, **it is a common practice in the current image generation field.**
>
> To further assess the perceptual loss incurred by using latent representations, we conducted the following experiment:
>
> **VAE:** Reconstruction quality of the input image using only the VAE encoder-decoder.
>
> **VAE + DDIM:** Standard reconstruction quality using the full diffusion model pipeline.
>
> **VAE + DDIM + VCA:** Reconstruction quality after incorporating our Visual Consistency Alignment (VCA) module.
>
> | Method |  CLIP Score↑ | NIMA AVA↑ | Avg ASR↓ |
> |--------|----------|---------|----------------|
> |   VAE    |  0.97           | 5.16        | 7.00       |
> | VAE+DDIM  | 0.89              | 5.03        |  22.10      |
> |   VAE+DDIM+VCA     | 0.97           | 5.13          | 7.60       |
>
> We found that the VAE itself introduces only a negligible perceptual degradation (with a 0.03 drop in CLIP score and a 0.17% increase in Avg ASR). Furthermore, we observed that **incorporating our VCA effectively compensates for the perceptual loss caused by denoising, reaching a level comparable to that of VAE compression.**
>
> Moreover, **another advantage of compressing into the latent space is that optimizing the latent-to-pixel mapping does not introduce noticeable noisy perturbations**, whereas direct optimization in the pixel space often does (as seen with DiffPGD in Figure 4).
>
> [1] Rombach R. et al., "High-resolution image synthesis with latent diffusion models," CVPR, 2022.
>
> ---
> **Weakness 4**
> *Details on LoRa Fine-Tuning, to be more precise, how LoRa modules are inserted.*
>
> Thank you for your rigorous review. As stated in Appendix F (lines 597–599), during the visual consistency alignment phase, we fine-tune only the projection matrices Q, K, and V in the UNet’s attention modules. We set the LoRA rank to 8 and train for 200 steps using the AdamW optimizer with a learning rate of 1×10⁻⁴. Before performing attack effectiveness alignment, we simply merge the LoRA parameters into the UNet, introducing no additional parameter overhead. For more implementation details, please refer to the source code included in the supplementary zip file.
>
> ---
> **Question 1**
> *In the paper, you conducted evaluation on existing defenses which originally focusing on Lp attacks, but are you also considering defenses method on unrestricted adversarial attacks?*
>
> Thank you for your careful feedback. The defense methods **Res-De (a model trained to reduce shape and texture bias) and Shape-Res (a model trained to increase shape bias for improved robustness)** in Table 2 are specifically designed for unrestricted adversarial examples, as introduced in line 273. Compared to prior work, our defense evaluation is more comprehensive, covering adversarial training (for both CNNs and ViTs), input preprocessing defenses, diffusion-based defenses, and defenses tailored to unrestricted attacks. However, as shown in Table 2, existing methods still struggle to defend against our strong adversarial examples, **which highlights a clear gap and motivates future research in this direction.**
>
> ---
> **Question 2**
> *In Eq.7, it introduces momentum-based gradients, but $\Pi_{z^0_T+\epsilon_a}$ , what this refer to? Can you provide more details?*
>
> In the adversarial attack domain, $\Pi_{z^0_T+\epsilon_a}$ is commonly defined as restricting the modification range of $z^0_T$ within $[ z^0_T-\epsilon_a, z^0_T+\epsilon_a ]$. Its purpose is to prevent the optimized $z_T$ from deviating too far from the original $z_T^0$. We will restate this more explicitly in the final version.
>
> ---
> **Question 4**
> *The “skip gradient” and “gradient checkpointing” strategies are described briefly; their impact on optimization stability and convergence, I think is not well justified or analyzed. Are there convergence guarantees ?*
>
> Thank you for your rigorous review. To ensure a fair comparison with prior work, we fixed the attack iteration number \( N \) to 10 steps and therefore did not include an explicit convergence analysis initially. Theoretically, gradient checkpointing yields more accurate gradient estimates than skip-gradient methods, which is supported by our experimental results showing that APA-GC outperforms APA-SG in attack effectiveness.
>
> We  supplement  a convergence analysis as follows:
> First, we use the white-box ASR as an indicator of convergence. In adversarial attacks, accurate gradients should result in a steadily increasing white-box ASR, reflecting effective optimization.
> Second, we randomly sampled 100 images and recorded the reward values over 15 iterations to observe convergence behavior. The results are summarized in the table below:
>
> | Method  | 2    | 4    | 6    | 8    | 10   | 12   | 14   | White-box ASR (%) |
> |---------|-------|-------|-------|-------|-------|-------|-------|-------------------|
> | APA-SG  | 5.32  | 7.19  | 9.10  | 10.86 | 12.27 | 12.94 | 13.16 | 99.6 |
> | APA-GC  | 6.67  | 10.27 | 13.39 | 16.20 | 18.81 | 19.51 | 20.31 | 99.7  |
>
> We observe a rapid increase in reward from steps 2 to 8, followed by slower growth from steps 10 to 14. Together with the high white-box ASR at step 10, this suggests the optimization essentially converges within 10 steps. Moreover, APA-GC shows a clearly faster reward increase than APA-SG. Due to certain methodological constraints, we cannot provide a clearer convergence curve at this stage, but we will include it in the final version of the paper.
>
> Once again, thank you for your time. We sincerely hope our response can address your concerns.

---

> > ### Comment · Reviewer_sWRo · 2025-08-06
> >
> > Thank you for the thorough and the extended empirical analysis.
> > As a follow-up, have you explored or do you have insights into specific scenarios where the latent similarity metric may not perfectly reflect human perceptual judgments, particularly in more semantically complex or fine-grained image settings? It would be interesting to see qualitative examples or a user study for cases where latent and pixel similarity diverge.
> >
> > I have read the concerns and comments from other reviewers. So, I have another question.  Have you considered or experimented with alternative insertion points or different LoRA ranks, and do you observe any trade-offs between attack success and visual fidelity depending on these choices? Ablation results could further illuminate this design choice.
> >
> > That's all from me. All the rest are cleared.

---

> > > ### Author Response · Authors · 2025-08-06
> > >
> > > Thank you for your detailed feedback. We would like to address your follow-up questions as follows:
> > >
> > >
> > >
> > > **A1:** This is an interesting question. As highlighted in the LDM paper, perceptual compression represents a trade-off between performance and efficiency. Consequently, a number of follow-up works have aimed to develop improved VAEs to reduce the information loss introduced by compression [1,2].
> > >
> > > To test your hypothesis, we analyze cases where the latent similarity is high but the pixel-level similarity is low. Since our method ensures a consistent magnitude of latent-space optimization across all images (as defined in Eq. 7), we can simply focus on reconstructed cases with low SSIM scores. Consistent with your intuition, we find that images with low SSIM values are predominantly high-frequency, detail-rich images, such as those containing text, skin, or fine textures.
> > >
> > > As illustrated in Figure 9, our method, when combined with ControlNet, improves the preservation of visual consistency in fine-grained images. We will include this analysis in the appendix by adding a figure showing the distribution of reconstruction quality and by providing representative qualitative examples. Thank you again for your insightful suggestion.
> > >
> > > [1] *Scaling Rectified Flow Transformers for High-Resolution Image Synthesis*, ICLR 2024.
> > > [2] *Deep Compression Autoencoder for Efficient High-Resolution Diffusion Models*, ICLR 2025.
> > >
> > > ---
> > >
> > >
> > >
> > > **A2:** Thank you for your suggestion. Regarding the LoRA insertion strategy, we followed the default configuration for LoRA finetuning on UNet. Additionally, we conducted experiments to analyze the impact of the LoRA rank on VCA performance. Using 50 randomly sampled images (source model: RN-50), we examined two aspects:
> > >
> > > **(1) Effect of rank on reconstruction quality (non-adversarial setting):**
> > >
> > > | Rank | CLIP Score ↑ | Avg ASR ↓ |
> > > |------|--------------|-----------|
> > > | 0    | 0.88         | 21.1      |
> > > | 4    | 0.97         | 7.2       |
> > > | 8    | 0.97         | 6.0       |
> > > | 16   | 0.97         | 5.0       |
> > >
> > > **(2) Effect of rank on performance under attack:**
> > >
> > > | Rank | CLIP Score ↑ | Avg ASR ↑ |
> > > |------|--------------|-----------|
> > > | 0    | 0.62         | 96.2      |
> > > | 4    | 0.87         | 79.2      |
> > > | 8    | 0.89         | 74.5      |
> > > | 16   | 0.91         | 71.2      |
> > >
> > > Overall, the results align with intuition: higher ranks lead to better visual consistency but come at the cost of reduced attack success rates and increased computational and training costs due to more tunable parameters. Considering this trade-off, we find that a rank of 8 provides a good balance. We will include these results and analyses in the appendix.
> > >
> > > ---
> > >
> > > Thank you once again for your thoughtful and constructive feedback. We hope our responses have addressed your concerns and helped to further improve our work. If these clarifications resolve your concerns, we would be grateful if you could consider raising your score.

---

> > > > ### Comment · Reviewer_sWRo · 2025-08-06
> > > >
> > > > Thank you for the detailed clarification. Now, all of my concerns have been cleared. I'm inclined to keep my score. Impressive work and wish you the best of luck.

---

> > > > > ### Author Response · Authors · 2025-08-07
> > > > >
> > > > > Thank you for your recognition and support of our work. We will carefully incorporate your suggestions into our final version.

---

### Official Review · Reviewer_a1PE · 2025-07-03

**Clarity:** 2
**Significance:** 3
**Originality:** 3
**Rating:** 4
**Confidence:** 4

**Summary:**

This paper proposes a two-stage framework APA (Adversary Preferences Alignment) to craft unrestricted adversarial examples using diffusion models. For an original image, APA first employs LoRA to fine-tune the UNet in the LDM to improve the reconstruction quality of DDIM inversion to ensure the visual consistency of the adversarial example after perturbing the latent embeddings. Then, during the denoising process of LDM, APA performs adversarial optimization on the latent embeddings through a dual-path strategy at both the trajectory-level and step-level. Additionally, APA introduces Diffusion Augmentation to further enhance the transferability.

**Questions:**

1. Except for the results in Figure  3, are there any quantitative results or theoretical analyses to support the effectiveness of Visual Consistency Alignment?
2. Considering the image quality of APA-SG in Figure 4 appears better than that of ACA and DiffPGD-MI, yet their quantitative metrics are comparable, and some results of APA-SG in Appendix H exhibit noticeable artifacts, are the visualization images specially selected? More visualization results of randomly sampled images should be included. Moreover, it is observed that APA-GC-P, which achieves better image quality according to quantitative metrics, suffers a significant drop in average ASR, but its detailed results are not included in Tables 1 and 2.
3. A comparison with the state-of-the-art LDM-based unrestricted adversarial attack DiffAttack [1] is missing. ([1] Chen J, Chen H, Chen K, et al. Diffusion models for imperceptible and transferable adversarial attack[J]. IEEE Transactions on Pattern Analysis and Machine Intelligence, 2024.)
4. It should be clarified whether the APA variant compared with AdvDiff in Appendix E.3 remains under an untargeted attack setting, to avoid potential unfair comparisons, or how APA can be extended to targeted attack scenarios.
5. Why does the optimization process of APA appear more complex and involve more computation steps than ACA, yet APA runs faster according to Appendix B? The authors should explain this result and provide a corresponding analysis for the computational complexity of APA.

**Ethical Concerns:**

["NO or VERY MINOR ethics concerns only"]

**Limitations:**

yes

**Quality:**

3

**Strengths And Weaknesses:**

The methodology presented in this paper demonstrates technical novelty. The APA approach is built upon the existing ACA method but incorporates several significant improvements. For example, it cleverly fine-tunes the UNet model parameters to enhance the visual consistency of Unrestricted Adversarial Examples. Additionally, it introduces both trajectory-level and step-level optimization of latent embeddings and diffusion augmentation to improve attack effectiveness.

However, the authors seem to overclaim the connection between APA and the concept of Performance Alignment in diffusion models. APA still follows the conventional paradigm of adversarial attacks, where each adversarial example must be generated through a complete gradient-based optimization process (for both UNet and latent embeddings in APA) with a given original image, rather than directly aligning the model once for solving this problem. Moreover, finding better solutions that simultaneously achieve high image quality and strong attack performance has long been a key objective in adversarial attacks. Although certain components of APA are defined as "rewards", they are essentially equivalent to the loss functions commonly used in existing adversarial attacks for computing gradients (e.g., cross-entropy loss from surrogate classification models).

---

> ### Author Rebuttal · Authors · 2025-07-30
>
> We sincerely thank the reviewer for their constructive feedback and for recognizing the technical contributions of our work.
> Below, we provide point-by-point responses to each comment.
>
> ---
>
> **Weakness 1:**
> *1. APA still follows per-sample gradient-based optimization and does not perform true model-level alignment.*
>
> *2. finding better solutions that simultaneously achieve high image quality and strong attack performance has long been a key objective in adversarial attacks.*
>
> *3. The so-called attack reward are essentially standard loss functions like cross-entropy.*
>
> Thank you for your comments. We would like to clarify that the key difference between APA and conventional attack optimization lies in its **alignment-driven design**. Our framework is motivated by how to:
> - define malicious preferences (*visual consistency* & *attack effectiveness*),
> - quantify them (*rule-based* or *learned rewards*),
> - select alignment strategies (*DPO*, *RL*, or *direct backpropagation*),
> - design task-specific modules (*dual-path optimization* & *diffusion augmentation*).
>
> These considerations lead to our proposed **two-stage alignment framework**. Below, we address the three concerns you raised in detail:
>
> 1. **Instance-level alignment**:
>    You're right, we perform *per-instance alignment*, not model-level alignment, as a practical choice due to the difficulty of generating unrestricted adversarial examples under conflicting preferences. This approach aligns with *test-time preference alignment* [1].
>
>     [1] Test-Time Preference Optimization: On-the-Fly Alignment via Iterative Textual Feedback, ICML,2025.
>
> 2. **No novelty claim on objective formulation**:
>    We do **not claim** that our objective design is novel. Instead, we extract preferences from adversarial attack literature. Our key contribution is to **decouple** the optimization of these preferences, showing clear advantages over joint optimization.
>
> 3. **Reward function is not our contribution**:
>    Under the direct backpropagation setting, it is indeed mathematically valid to formalize a reward function as a loss. Actually, reward design is not our contribution (discussed in Section 4.2). Our key contributions under this setting lie in **dual-path attack optimization** and **diffusion augmentation**, which are specifically designed to address known issues with direct backpropagation (such as overfitting to the reward model).
>
> In summary, we believe the **technical similarity between alignment via backpropagation** and **gradient-based optimization in adversarial attacks** may have led to some misunderstanding regarding the novelty of our work. However, the core innovation of APA is the decoupled alignment of two preferences (instead of joint optimization) and the design of task-specific modules that effectively mitigate overfitting issues in direct backpropagation settings. We further support our claims through extensive experiments demonstrating the superiority, flexibility, and robustness of our method.
>
> ---
>
>  **Question 1:**
> *Except for the results in Figure 3, are there any quantitative results or theoretical analyses to support the effectiveness of Visual Consistency Alignment?*
>
> We sincerely appreciate the reviewer’s careful analysis. To further clarify the role of our proposed **Visual Consistency Alignment (VCA)**, we have conducted two additional ablation studies.
>
> 1. To evaluate whether VCA improves the inherent reconstruction ability of diffusion models, we compare DDIM with and without VCA under a non-attacking setting.
>
>     | Method | LPIPS↓ | SSIM↑ | CLIP Score↑ | NIMA AVA↑ | CNNIQA↑ | Avg ASR↓ |
>     |--------|----------|---------|----------------|-------------|-----------|------------|
>     | DDIM| 0.19| 0.73| 0.89| 5.03| 0.63| 22.10|
>     |DDIM+VCA| 0.05| 0.85| 0.97| 5.13| 0.63| 7.60|
>
> These results suggest that **VCA clearly enhances the reconstruct quality** of diffusion model.
>
> 2. We also evaluated the quality of images generated when only the attack effectiveness alignment (i.e., without VCA):
>     | Method | LPIPS↓ | SSIM↑ | CLIP Score↑ |
>     |--------------------|----------|---------|----------------|
>     | APA-SG w/o VCA | 0.55  | 0.46  | 0.62   |
>     | APA-SG  | 0.25 | 0.67 | 0.86 |
>
> The noticeable degradation in visual quality upon the removal of VCA highlights **its essential contribution to maintaining visual consistency during attack alignment.**
> We will incorporate all the above findings into the revised paper to better communicate the role and advantages of VCA.
>
> ---
>
>  **Question 2.1:**
> *APA-SG’s images in Figure 4 look better than ACA and DiffPGD-MI despite similar metrics, yet Appendix H shows artifacts. Were the visuals selectively chosen? Please provide more random samples.*
>
>
> Thank you for your careful observation. First, we would like to clarify that **we did not deliberately select qualitative examples**. Due to the constraints, we could not include the full set of test results. However, you can reproduce them using the provided code in the supplementary materials. We will release all results, including those of the baselines, upon acceptance of the paper.
>
> In addition, **we believe that our quantitative analysis is consistent with the qualitative findings.** For instance, DiffPGD-MI introduces noticeable background perturbations while preserving image structure, leading to high SSIM but low aesthetics. ACA's low CLIP score reflects its tendency to alter image semantics. In contrast, APA-SG mainly modifies the background while preserving the subject, producing more natural outputs and thus a higher aesthetic score.
>
> ---
>
>  **Question2.2 and Question3:**
> *1. Could the authors clarify why the detailed results of APA-GC-P, which appear to offer better image quality but lower ASR, are not included in Tables 1 and 2?*
>
> *2. Comparison with the state-of-the-art LDM-based attack DiffAttack [1] seems to be missing."*
>
> Thank you for your valuable suggestions. We have added detailed results for **APA-GC-P** and **DiffAttack**.
> For (a) and (b), we report the **Avg. ASR** due to space limitations. The complete results will be included in the final version.
>
> **a. Attack Performance Comparison (Table 1)**
>
> | Source model  | DiffAttack | APA-SG | APA-GC | APA-GC-P |
> | --- | --- | --- | --- | --- |
> | MobViT-s | 70.02 | 65.85 | 77.48 | 52.56 |
> | MN-v2 | 59.05 | 72.14 | 87.01 | 56.67 |
> | RN-50 | 65.10 | 75.32 | 88.02 | 62.08 |
> | ViT-B | 45.06 | 66.21 | 74.82 | 45.90 |
> | Average | 59.79 | 69.88 | 81.83 | 54.30 |
>
> **b. Attacks on Adversarial Defense (Table 2)**
> | Attacks | Avg ASR |
> | --- | --- |
> | DiffAttack | 37.17 |
> | APA-SG | 63.59 |
> | APA-GC | 70.20 |
> | APA-GC-P | 39.26 |
>
> **c.  Visual Quality Comparison (Table 3)**
>
> | Method |  LPIPS↓ | SSIM↑ | CLIP Score↑ | NIMA AVA↑ | CNNIQA↑ | Avg ASR↑ |
> | --- | --- | --- | --- | --- | --- | --- |
> | DiffAttack | 0.14 | 0.68 | 0.87 | 5.17 | 0.66 | 65.10 |
> | APA-SG | 0.25 | 0.67 | 0.86 | 5.29 | 0.62 | 75.32 |
> | APA-GC | 0.23 | 0.69 | 0.83 | 5.39 | 0.67 | 88.02 |
> | APA-GC-P | 0.09 | 0.82 | 0.91 | 5.22 | 0.63 | 62.08 |
>
> **Conclusion:**
>
> 1. Our APA-SG and APA-GC **achieve 10.09 and 22.04 higher attack performance than DiffAttack, respectively.**
>
> 2. Under defense scenarios, **APA-GC, APA-SG, and APA-GC-P all demonstrate better robustness compared to DiffAttack.**
>
> 3. APA-GC achieves comparable visual similarity and aesthetic scores to DiffAttack. Moreover, **APA-GC-P exhibits significantly better visual quality than DiffAttack with comparable attack performance.** We will include qualitative examples in the main paper as well (we observed that images generated by DiffAttack often have overly sharp object boundaries).
>
> 4. **These results further demonstrate the superiority of our two-stage design in APA over the joint optimization used in DiffAttack,** enabling more effective and controllable generation.
>
> 5. As noted, APA-GC-P offers superior visual quality but moderate attack effectiveness. We include it to showcase the flexibility of our framework in supporting task-specific optimization choices.
>
> ---
>
> **Question4:**
> *It should be clarified whether the APA variant compared with AdvDiff in Appendix E.3 remains under an untargeted attack setting, to avoid potential unfair comparisons, or how APA can be extended to targeted attack scenarios.*
>
> Thank you for your insightful comments. As stated in its original paper('Evaluation metrics' on page 10), both AdvDiff and AdvDiff-untarget (which enhances transferability) are intended for evaluating **untargeted attacks.** Accordingly, **all comparisons were performed under the untargeted setting, with fairness across all baselines strictly maintained.** Moreover, extending APA to targeted attacks is straightforward—one only needs to replace the optimization objective accordingly. Please refer to lines 612–619 for implementation details.
>
> ---
>
> **Question5:**
>
> *APA appears more complex and involves more computation steps than ACA, yet runs faster according to Appendix B*
>
> Thank you for your suggestion. We identified two contributing factors. First, **our VCA optimization is significantly faster than APA’s null text optimization (e.g., 38 s vs. 55 s)**. Second, our Stage-2 alignment is implemented on a lightweight codebase based on Diffusers, whereas ACA relies on a heavier framework, which introduces slight overhead. You may verify this using the code provided in the supplementary zip. **Our codebase is clean and modular—easily integrable into Diffusers, with flexible parameter selection and customizable optimization strategies.**
>
> Regarding the complexity analysis:
> Stage 1 has a time complexity of **O(n)** (number of training steps),
> while Stage 2 is **O(N·T)** (attack iterations × denoising steps).
>
> Once again, thank you for your time. We sincerely hope our response can address your concerns.

---

> > ### Comment · Reviewer_a1PE · 2025-08-09
> >
> > Thank you to the authors for their detailed response. The authors have addressed most of my concerns, and I have also read the comments from other reviewers. Although I still feel that the original paper contains some overclaims in its writing, given that performance alignment is indeed one of the current research hotspots, the contributions remain sufficient. Therefore, I will maintain my rating as 4, and I hope the authors can try to eliminate some of the overclaims and ambiguities related to the concept of preference alignment in the revised version.

---

> > > ### Author Response · Authors · 2025-08-09
> > >
> > > Thank you for your feedback. We greatly appreciate your recognition of the technical innovations and performance contributions of our work, which is consistent with the feedback provided by other reviewers. In the final version, we will address the writing concern you mentioned. Specifically, we will revise the introduction and Section 4—building on our response to Weakness 1 and your suggestions—to more precisely describe how we draw methodological inspiration from preference alignment techniques and adapt them to the adversarial attack setting. In addition, we will further elaborate in the second paragraph of the Related Work section on how our APA framework relates to and differs from conventional attack optimization approaches. Once again, thank you for your valuable input and your positive response to our paper.

---

### Note · Authors · 2025-08-11

We sincerely thank all reviewers for their efforts and recognition. The feedback has been invaluable in improving our paper. We briefly summarize the main points of discussion here.

Our paper presents a novel perspective framing unrestricted adversarial example generation as adversary preferences alignment. We propose two-stage alignment framework APA, which addresses optimization instability by decoupling conflicting preferences (attack effectiveness vs. visual consistency) . Experimental results show APA significantly outperforms joint optimization. Reviewers provide positive feedback, with a1PE noting "technical novelty" and sWRo highlighting the value of "Concept of aligning diffusion models to adversary preferences rather than benign human preferences". Our contributions are recognized by all reviewers as valuable to the community.

Through reviewer discussions, we further refine our paper. We enhance experiments by adding DiffAttack comparisons (a1PE), quantitative visual consistency analysis (a1PE, sWRo, KeyC), efficiency evaluations (a1PE, ELVx, KeyC), and convergence studies (sWRo). We improve writing by clarifying APA's distinctions from traditional attack optimization (a1PE, ELVx), adding symbol tables (sWRo), adopting an inverse problem sampling perspective (ELVx), and justifying image-specific optimization rationale (KeyC).

Additionally, inspired by reviewers, we reinforce three important findings:

1. **Two-stage superiority:**  In discussions  with a1PE further validated that our two-stage APA framework significantly outperforms joint optimization methods (e.g. ACA, DiffAttack) .

2. **Further efficiency optimization:** In discussions with ELVx and sWRo, we discovered that our APA exhibits rapid convergence trends, allowing further efficiency improvements through reduced attack iterations and denoising optimization steps while maintaining leading attack performance.

3. **Video extensibility:** Inspired by KeyC, our LoRA approach for visual consistency shows good generalization between video frames, suggesting efficient extension to video adversarial generation.

We once again thank all reviewers and the AC for their support and assistance with our work. We are delighted to witness such constructive academic exchange. We have ensured that all additional analyses will appear in the final version and will release all code and data mentioned during the response process.

---

### Decision · Program_Chairs · 2025-09-17

**Decision:**

Accept (poster)

**Comment:**

This paper proposes a diffusion-based method for crafting unrestricted adversarial attacks that aims to balance adversarial performance and consistency. Reviewers found the paper to make a meaningful technical contribution to the field. The discussion was very fruitful and helped identify several clarity issues in the current draft - acceptance is conditioned on the authors incorporating all points raised by the reviewers in their next revision. One question raised was wether the connection to preference alignment is needed or useful. Here the authors should decide weather to make it less pronounce, or to adopt it fully (in which case the related works section should be updated, e.g. see [1]). The authors should pay special attentions to the concerns raised regarding ambiguities, overclaims, and portraying the method's limitations.

[1] Adversaries With Incentives: A Strategic Alternative to Adversarial Robustness, ICLR 2025.